# Fluorescence activation mechanism and imaging of drug permeation with new sensors for smoking-cessation ligands

Aaron L Nichols[1,†], Zack Blumenfeld[1,2,†], Chengcheng Fan[3,†], Laura Luebbert[1,4,†], Annet EM Blom[3], Bruce N Cohen[1], Jonathan S Marvin[5], Philip M Borden[5], Charlene H Kim[1], Anand K Muthusamy[3], Amol V Shivange[1], Hailey J Knox[3], Hugo Rego Campello[6], Jonathan H Wang[1], Dennis A Dougherty[3], Loren L Looger[5], Timothy Gallagher[6], Douglas C Rees[3,7], Henry A Lester[1]*

[1]Division of Biology and Biological Engineering, California Institute of Technology, Pasadena, United States; [2]Keck School of Medicine, University of Southern California, Los Angeles, United States; [3]Division of Chemistry and Chemical Engineering, California Institute of Technology, Pasadena, United States; [4]Institute of Biology, Leiden University, Leiden, Netherlands; [5]Janelia Research Campus, Howard Hughes Medical Institute, Ashburn, United States; [6]School of Chemistry, University of Bristol, Bristol, United Kingdom; [7]Howard Hughes Medical Institute, California Institute of Technology, Pasadena, United States

*For correspondence:
lester@caltech.edu

†These authors contributed equally to this work

Competing interest: The authors declare that no competing interests exist.

**Abstract** Nicotinic partial agonists provide an accepted aid for smoking cessation and thus contribute to decreasing tobacco-related disease. Improved drugs constitute a continued area of study. However, there remains no reductionist method to examine the cellular and subcellular pharmacokinetic properties of these compounds in living cells. Here, we developed new intensity-based drug-sensing fluorescent reporters (iDrugSnFRs) for the nicotinic partial agonists dianicline, cytisine, and two cytisine derivatives – 10-fluorocytisine and 9-bromo-10-ethylcytisine. We report the first atomic-scale structures of liganded periplasmic binding protein-based biosensors, accelerating development of iDrugSnFRs and also explaining the activation mechanism. The nicotinic iDrugSnFRs detect their drug partners in solution, as well as at the plasma membrane (PM) and in the endoplasmic reticulum (ER) of cell lines and mouse hippocampal neurons. At the PM, the speed of solution changes limits the growth and decay rates of the fluorescence response in almost all cases. In contrast, we found that rates of membrane crossing differ among these nicotinic drugs by >30-fold. The new nicotinic iDrugSnFRs provide insight into the real-time pharmacokinetic properties of nicotinic agonists and provide a methodology whereby iDrugSnFRs can inform both pharmaceutical neuroscience and addiction neuroscience.

## Editor's evaluation

Nichols et al. developed and characterized the first fluorescent sensors for several nicotinic receptor partial agonists relevant to smoking cessation. It is potentially a major advance for the field. They leveraged crystallography to understand the mechanism by which the ligands enhance fluorescence, then characterized top sensors for sensitivity, selectivity, and kinetics, and their utility in plasma membrane and ER sensing in neurons and cell lines. The tools developed by this team will enable investigators to track nicotinic receptor partial agonists in different subcellular compartments with relatively fast time resolution.

## Introduction

Smoking cessation is an important goal to help decrease the burden, both individual and societal, of tobacco-related disease. The addictive tobacco alkaloid nicotine itself, via transdermal patches and other devices, remains available for people trying to quit smoking; but nicotine replacement therapy has distressingly low rates of success. Therefore, various research projects are continuing with the aim of developing more effective ligands for nicotinic acetylcholine receptors (nAChRs).

Prior work suggests that partial agonists with lower efficacy than nicotine could serve as effective smoking-cessation drugs (*Rose et al., 1994*), and efforts continue in that direction (*Rollema and Hurst, 2018*). Another plant alkaloid, (-)-cytisine (also called cytisinicline and Tabex), an α4β2 nAChR partial agonist, served as a basis for the synthesis of analogs that have not yet entered the clinic (*Chellappan et al., 2006*; *Houllier et al., 2006*; *Imming et al., 2001*; *Kozikowski et al., 2007*; *Marcaurelle et al., 2009*; *Philipova et al., 2015*; *Rouden et al., 2002*). Varenicline (Chantix) has four rings, two more than nicotine or cytisine, and is currently the only FDA-approved smoking-cessation drug, but the modest quit rate of ~18% at 12 months invites further investigation (*Coe et al., 2005*; *Mills et al., 2009*). Dianicline, another tetracyclic compound, was discontinued after unfavorable Phase III clinical trials (*Cohen et al., 2003*; *Fagerstrom and Balfour, 2006*).

A nicotinic ligand for smoking cessation must satisfy at least three criteria (*Rollema et al., 2010*; *Tashkin, 2015*). (1) It must enter the brain, where the most nicotine-sensitive nAChRs (α4β2) occur. It must also (2) activate α4β2 nAChRs with an $EC_{50}$ sufficient to reduce cravings and withdrawal (1–2 µM). Finally, it must (3) block nicotine binding to reduce the reward phase of smoking (2–30 min). Varenicline meets these criteria, while cytisine (low brain penetration) and dianicline ($EC_{50}$ = 18 µM) each fail one of the criteria (*Rollema et al., 2010*).

Membrane permeation is interesting for investigating and treating nicotine addiction in at least two ways. Firstly, note criterion #1 above. For uncharged molecules, the conventional metric for membrane permeability is logP, where P is the octanol-water partition coefficient. For weak bases including most orally available neural drugs, logP must be corrected to account for the fraction of uncharged (deprotonated) molecules at the pH of interest, usually pH 7.4; the resulting metric, termed $logD_{pH7.4}$, is always less positive than logP. Enhancing the membrane permeability of cytisine analogs and probing nAChR subtype selectivity was addressed via direct functionalization of cytisine within the pyridone ring (*Rego Campello et al., 2018*). Two of the resulting derivatives, 10-fluorocytisine and 9-bromo-10-ethylcytisine, have cytisine-like $EC_{50}$ for the α4β2 nAChRs, but more positive calculated $logD_{pH7.4}$ values, suggesting greater membrane permeability at the nearly neutral pH of the blood, brain, and cytoplasm (*Blom et al., 2019*). Estimates of $logD_{pH7.4}$ are inexact, extrapolated, or rely on algorithmic calculations whose results differ over 2 log units for individual molecules (*Pieńko et al., 2016*). These estimates have unknown applicability to biological membranes at the $logD_{pH7.4}$ values < 0 that characterize varenicline, dianicline, and the cytisine analogs.

Secondly, nicotine dependence involves one or more 'inside-out' mechanisms. Nicotine itself ($logD_{pH7.4}$ 0.99) enters the endoplasmic reticulum (ER), binds to nascent nAChRs, becomes a pharmacological chaperone for the nAChRs, and eventually causes selective upregulation of these receptors on the plasma membrane (PM) (*Henderson and Lester, 2015*). For this reason, it is especially important to understand permeation into the ER.

These two neuroscience aspects of nicotinic ligands – pharmaceutical science and addiction science – call for direct measurements of drug movements in living cells (*Video 1*). We previously explored the subcellular pharmacokinetics of nicotine and varenicline in immortalized cell lines and cultured neurons using the iDrugSnFRs iNicSnFR3a and iNicSnFR3b to visualize that these nicotinic agonists enter the ER within seconds of drug application and exit equally rapidly from the ER upon extracellular washing (*Shivange et al., 2019*). That nicotine diffuses across cellular membranes in seconds has been suspected for decades: nicotine crosses six PMs to enter the brain within 20 s, providing a 'buzz.'

**Video 1.** Genetically encoded fluorescent biosensors show how drugs cross membranes in real time.
https://elifesciences.org/articles/74648/figures#video1

That varenicline becomes trapped in acidic vesicles suggests appreciable membrane permeation but may also underlie unwanted effects (*Govind et al., 2017*; *Le Houezec, 2003*).

We sought to generate and apply additional intensity-based drug-sensing fluorescent reporters (iDrugSnFRs) for candidate smoking-cessation drugs: dianicline, cytisine, 10-fluorocytisine, and 9-bromo-10-ethylcytisine. We hypothesized that a family of newly developed iDrugSnFRs would enable quantifiable fluorescence signals that compare the differences in permeation among these compounds.

## Results

### Generation of additional nicotinic iDrugSnFRs: Structural tactic

To generate iDrugSnFRs for cytisine and dianicline, we followed two converging tactics. In the 'structure-based' tactic, we obtained the first structural data for OpuBC-based SnFRs bound by nicotinic ligands (nicotine and varenicline) (*Figure 1*, *Supplementary file 1*). Crystals of iNicSnFR3adt in the presence of 10 mM nicotine diffracted to 2.95 Å resolution (PDB 7S7U). Overall, the liganded periplasmic binding protein (PBP) domain of iNicSnFR3adt adopts a closed conformation (*Figure 1A*). In the binding pocket between the top and bottom lobes of the PBP, we observed an 'avocado'-shaped electron density in the nicotine binding site, enclosed by several aromatic residues (*Figure 1B*). The combination of protonation/deprotonation and the rotatable bond of nicotine (*Elmore and Dougherty, 2000*) vitiate unambiguously localizing it within the binding pocket.

We obtained an unambiguous ligand placement for iNicSnFR3adt in the presence of 10 mM varenicline in the same crystallization condition. Crystals of iNicSnFR3adt with varenicline bound were isomorphous to those of the nicotine-bound crystals and diffracted to 3.2 Å resolution (PDB 7S7T). While the protein structure (*Figure 1D*) is identical to that of the nicotine bound structure (*Figure 1A*), the rigidity and additional ring of varenicline allowed us to unambiguously localize it in the binding pocket. Varenicline is enclosed by the same aromatic residues as nicotine, forming cation-π interactions with Tyr65 and Tyr357, in addition to other interactions with the pocket residues (*Figure 1E*).

The data confirm that similar ligand-induced conformational changes occur in the PBP for nicotine, varenicline, ACh (*Borden et al., 2019*), and choline (*Fan, 2020*; *Figure 1—figure supplement 1*). These changes resemble those for other OpuBC PBPs (*Schiefner et al., 2004*).

In the full iDrugSnFR, in the apo state, the Glu78 in Linker 1 approaches within ~2.5 Å of the oxygen of the tyrosine fluorophore (*Figure 1E1*; PDB 7S7V). *Figure 1E2* provides structural details confirming the hypothesis (*Barnett et al., 2017*; *Nasu et al., 2021*) that in the liganded state Glu78 has moved away, presumably allowing the fluorescent tyrosinate to form (*Video 2*). We term this mechanism the 'candle snuffer'.

### Generation of additional nicotinic iDrugSnFRs: Mutational tactic

In the mutational tactic, we screened each drug shown in *Figure 1—figure supplement 2* against a panel of biosensors that included iNicSnFR3a and iNicSnFR3b (*Shivange et al., 2019*) and iAChSnFR (*Borden et al., 2019*) as well as intermediate constructs from their development process. From this screen, we chose sensors with the lowest $EC_{50}$ for each drug as our starting protein for iDrugSnFR evolution.

Because the candle snuffer mechanism explains several details of the agonist- and pH sensitivity of both iNicSnFR3a and iSketSnFR (see 'Discussion'), we presume that it represents a general mechanism for OpuBC-cpGFP SnFRs. We did not mutate residues that lie (in 3D space) between the binding site and linkers.

For dianicline and cytisine separately, we incrementally applied site-saturation mutagenesis (SSM) to first- and second-shell amino acid positions within the binding pocket. We evaluated each biosensor and drug partner in lysate from *Escherichia coli* and carried forward the biosensor with the highest S-slope to the subsequent round. S-slope, $\frac{\frac{\Delta F}{F_0}}{[ligand]}$ at the beginning of the dose–response relation, emphasizes the response to ligand concentrations in the pharmacologically relevant range (*Bera et al., 2019*). *Table 1* and *Figure 2* summarize dose–response relations for the optimized sensors. The dianicline sensor, iDianiSnFR, has $EC_{50}$ 6.7 ± 0.3 µM, $\Delta F_{max}/F_0$ 7.4 ± 0.1, and S-slope 1.1. The cytisine sensor, iCytSnFR, has $EC_{50}$ 9.4 ± 0.8 µM, $\Delta F_{max}/F_0$ 5.0 ± 0.2, and S-slope 0.5 (*Table 1*, *Figure 2A and B*). After generating iCytSnFR, we performed additional SSM to progress from iCytSnFR to SnFRs for

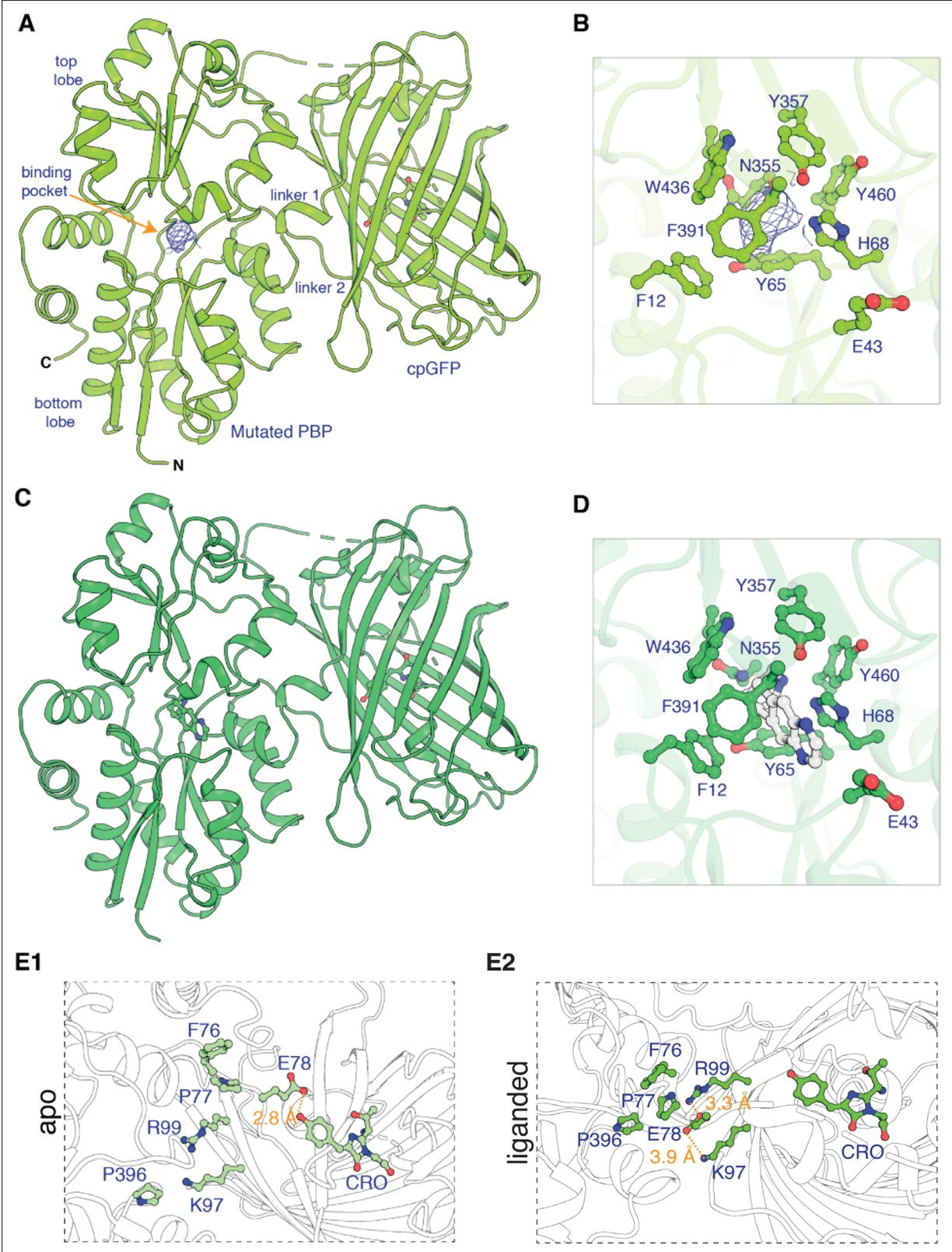

**Figure 1.** Apo and ligand-bound structures of iNicSnFR3adt (dt indicates that His$_6$ and Myc tags have been removed to aid crystallization). To form an intensity-based drug-sensing fluorescent reporter (iDrugSnFR), a circularly permuted GFP molecule, flanked by two 4-residue linking sequences, is inserted into a PBP at a position (77–78, in our numbering system) that changes backbone Φ-Ψ angles between the apo and liganded PBP. (**A**) Overall conformation of iNicSnFR3adt crystallized with nicotine; an electron density appears at the nicotine binding site (PDB 7S7U) (**B**) iNicSnFR3adt binding

*Figure 1 continued on next page*

*Figure 1 continued*

site residues. (**C**) Overall conformation of iNicSnFR3adt with varenicline bound (PDB 7S7T). (**D**) iNicSnFR3adt binding site with varenicline present. (**E**) Aspects of the PBP-Linker1-cpGFP interface, emphasizing contacts that change upon ligand binding. The Phe76-Pro77-Glu78 cluster (in Linker 1) lies 11–16 Å from position 43, which defines the outer rim of the ligand site (**B**); therefore, the cluster makes no direct contact with the ligand site. (**E1**) In the apo conformation, Glu78 acts as a candle snuffer that prevents fluorescence by the chromophore (PDB 7S7V). (**E2**) In the liganded conformation (PDB 7S7T), the Phe76-Pro77-Glu78 cluster moves Glu78 at least 14 Å away from the fluorophore. Pro77 is flanked by Phe76 and Pro396 (in the top lobe of the PBP moiety). The presumably deprotonated Glu78 forms salt bridges with Lys97 and Arg99, both facing outward on the β6 strand of the original GFP (within the original Phe165-Lys-Ile-Arg-His sequence).

The online version of this article includes the following figure supplement(s) for figure 1:

**Figure supplement 1.** Conformational change of apo (PDB 7S7V) to the liganded, closed form (PDB 7S7T) of iNicSnFR3adt.

**Figure supplement 2.** Left: electrostatic surface potential densities for protonated forms of the nicotinic agonists in this study, calculated by SPARTAN at HF/6-31G** theory level.

10-fluorocytisine and 9-bromo-10-ethylcytisine. This optimization gave us iCyt_F_SnFR ($EC_{50}$ 1.4 ± 0.04 µM, $\Delta F_{max}/F_0$ 7.9 ± 0.1, S-slope 5.6) and iCyt_BrEt_SnFR ($EC_{50}$ 5.7 ± 0.1 µM, $\Delta F_{max}/F_0$ 4.0 ± 0.03, and S-slope 0.7) (*Table 1*, *Figure 2C and D*).

## Specificity and thermodynamics of nicotinic iDrugSnFRs

We characterized the specificity of purified iDrugSnFRs for their drug partners versus a panel of related nicotinic agonists (*Table 2*, *Figure 3*). The newly developed iDrugSnFRs showed some sensitivity to related nicotinic agonists. iDianiSnFR had the greatest fidelity for its drug partner but also showed an increased $EC_{50}$ (15 µM) as a purified protein versus its $EC_{50}$ in lysate (6.7 µM), possibly indicating decreased stability in a purified form. iCytSnFR, iCyt_F_SnFR, and iCyt_BrEt_SnFR showed a greater level of promiscuity for the compounds comprising the nicotinic agonist panel. Of note, iCytSnFR, iCyt_F_SnFR, and iCyt_BrEt_SnFR have an exceptionally low (60–90 nM) $EC_{50}$ for varenicline. The newly developed iDrugSnFRs showed negligible binding to choline or the neurotransmitter acetylcholine, leading one to expect minimal endogenous interference during future in vivo experiments.

We also performed dose–response experiments with iDianiSnFR, iCytSnFR, iCyt_F_SnFR, and iCyt_BrEt_SnFR against a panel of nine endogenous molecules, including neurotransmitters (*Figure 3—figure supplement 1*). iDianiSnFR showed no response to any of the nine selected compounds above background. iCytSnFR, iCyt_F_SnFR, and iCyt_BrEt_SnFR showed no response above background for seven of the compounds. However, they exhibited a $\Delta F/F_0$ of 0.25–0.8 to dopamine at 316 µM/1 mM and a $\Delta F/F_0$ of 0.8–1.5 to serotonin (5-HT) at 316 µM/1 mM. In terms of S-slope, the relevant metric for most cellular or in vivo experiments, the SnFRs are at least 250-fold more sensitive to their eponymous partners than to other molecules we have tested.

To examine the thermodynamics of the iDrugSnFR:drug interaction, we conducted isothermal titration calorimetry (ITC) binding experiments (*Figure 4*). The experimentally determined $K_D$ of each iDrugSnFR:drug pair using ITC was within a factor of 1.5 from the experimentally determined $EC_{50}$ for fluorescence in *E. coli* lysate or purified protein (*Table 3*). We infer that the $EC_{50}$ for fluorescence is dominated by the overall binding of the ligand for all the iDrugSnFRs.

## Kinetics of nicotinic agonist iDrugSnFRs: Stopped-flow

In a stopped-flow apparatus, we measured the fluorescence changes of iDrugSnFRs with millisecond resolution during multiple 1 s trials and an independent 100 s trial. The stopped-flow data revealed that iDrugSnFRs do not have pseudo-first-order kinetic behaviors typical of two-state binding interactions. Time courses of iDianiSnFR (both over 1 s and 100 s) were best fitted by double exponential equations. Most of the fluorescence change occurs within the first 0.1 s of mixing (*Figure 5A*), with only minor additional increase by 100 s.

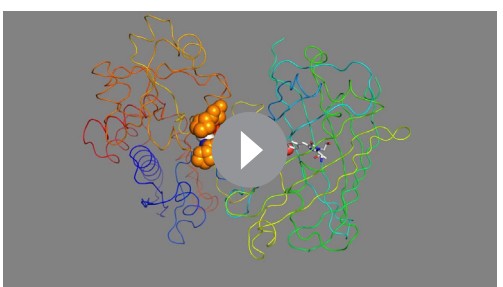

**Video 2.** Video morph of PDB 7S7V to 7S7T. PBP at the left; cpGFP at the right; key side chains in the linkers are shown as spheres. The ligand, varenicline, is shown as sticks.

https://elifesciences.org/articles/74648/figures#video2

**Table 1.** Nicotinic agonist iDrugSnFR naming, dose-response relations, and residues mutated. Parent constructs in bold. Measurements in *E. coli* lysates (L) or with purified protein (P). ND, not determined. Data for iAChSnFR from *Borden et al., 2019*; data for iNicSnFR3b from *Shivange et al., 2019*.

| Informal name | Drug of interest | ΔF$_{max}$/F$_0$ | | EC$_{50}$ (μM) | | S-slope | | Residues mutated vs. parent constructs | | | | | | | |
|---|---|---|---|---|---|---|---|---|---|---|---|---|---|---|---|
| | | L | P | L | P | L | P | 11 | 43 | 44 | 68 | 324 | 360 | 391 | 395 |
| **iNicSnFR3b** | Nicotine | ND | 10 | ND | 19 | ND | 0.5 | E | E | N | H | S | T | F | G |
| iDianiSnFR | Dianicline | 7.4 ± 0.1 | 4.7 ± 0.2 | 6.7 ± 0.3 | 15 ± 1 | 1.1 | 0.3 | D | R | - | S | N | G | - | N |
| **iAChSnFR** | ACh | ND | 12 | ND | 1.3 | ND | 9.2 | I | V | N | H | A | T | F | G |
| iCytSnFR | Cytisine | 5.0 ± 0.2 | 7.3 ± 0.4 | 9.4 ± 0.8 | 11 ± 1 | 0.5 | 0.7 | - | Y | - | - | - | - | W | - |
| iCyt_F_SnFR | 10-Fluorocytisine | 7.9 ± 0.1 | 2.3 ± 0.1 | 1.4 ± 0.04 | 1.6 ± 0.3 | 5.6 | 1.4 | - | N | G | - | - | - | W | - |
| iCyt_BrEt_SnFR | 9-Bromo-10-ethylcytisine | 4.0 ± 0.03 | 3.6 ± 0.04 | 5.7 ± 0.1 | 4.2 ± 0.2 | 0.7 | 0.9 | - | Q | G | - | - | - | W | - |

Changes in fluorescence from iCytSnFR during the first 1 s of mixing fit well to a single exponential (*Figure 5B*) and have close to pseudo-first-order kinetics (i.e., the observed rate of fluorescence change is nearly linear with drug concentration). As with iDianiSnFR, most of the fluorescence change occurs within the first second, with additional fluorescent increase continuing over the next minute (*Figure 5B*, right panel).

Like iDianiSnFR, iCyt_F_SnFR fluorescence changes are best fit by a double exponential (*Figure 5C*), but the time course of fluorescence change is significantly slower. Fluorescence gradually increases throughout the recording period and beyond. This information was considered in later in vitro and ex vivo experiments. iCyt_BrEt_SnFR fits well to a single exponential (*Figure 5D*) for the first 1 s of data collection, but like the other sensors, continues to increase its fluorescence over longer periods.

We plotted the k$_{obs}$ (s$^{-1}$) obtained in the 1 s stopped-flow experiments versus concentration (*Figure 5E–H*) (see also *Supplementary file 2*). The aberrations from ideal first-order kinetics vitiate generation of definitive k$_{off}$ and k$_{on}$ values but we can approximate a K$_{max}$ and K$_D$ from our fitting procedures. Our stopped-flow experiments reinforced previous observations (*Unger et al., 2020*) that the kinetics of iDrugSnFR binding involve complexities beyond a simple first-order kinetic model governing two binding partners.

## Kinetics of nicotinic agonist iDrugSnFRs: Millisecond microperfusion

We also studied iCytSnFR_PM expressed in HEK293T cells during fluorescence responses to ACh, cytisine, or varenicline in a microperfusion apparatus that exchanged solutions near the cell on a millisecond time scale (Materials and methods). This system directly measures the decay of the response when ligand is suddenly removed. The rank order of the iCytSnFR steady-state sensitivities is varenicline > cytisine > ACh. The time constant for decay decreased with increasing steady-state EC$_{50}$ of the ligands, as though more tightly binding ligands dissociate more slowly (*Figure 6A*).

We measured the decay waveforms after drug pulses at concentrations ≥ the EC$_{50}$ of the steady-state response to maximize the ΔF/F$_0$ signal/noise ratio (*Figure 6A–D*). Because the decay phases are measured in zero [ligand], one expects that the decay rate constant(s) (k$_{off}$) for an iDrugSnFR do not depend on the pulsed ligand concentration. Decay of the ACh response followed a single exponential time course (*Figure 6B*). The values of the k$_{off}$ for 30, 100, and 200 μM ACh did not differ significantly (ANOVA, p=0.62, degrees of freedom [df] = 2 (model), 20 [error]). We pooled them to obtain a mean k$_{off}$ of 1.9 ± 0.1 s$^{-1}$ (mean ± SEM, n = 23 areas [50 cells]). The corresponding time constant $\tau_{off}$ was 530 ± 30 ms. Hence, the temporal resolution of the CytSnFR_PM sensor for changes in the ACh concentration was in the subsecond range.

The decay of the cytisine and varenicline response was biphasic (*Figure 6C and D*): two exponential decay terms with an additional constant component fitted the cytisine decay significantly better than a single exponential term (F-test, p<0.05). As expected, neither the faster decay rate constants (kf$_{off}$) (ANOVA, p=0.30, df = 3,32) nor the slower decay rate constants (ks$_{off}$) (ANOVA, p=0.54, df = 3,31) differed among the tested cytisine concentrations (5–15 μM). The kf$_{off}$ and ks$_{off}$ for 5–15 μM

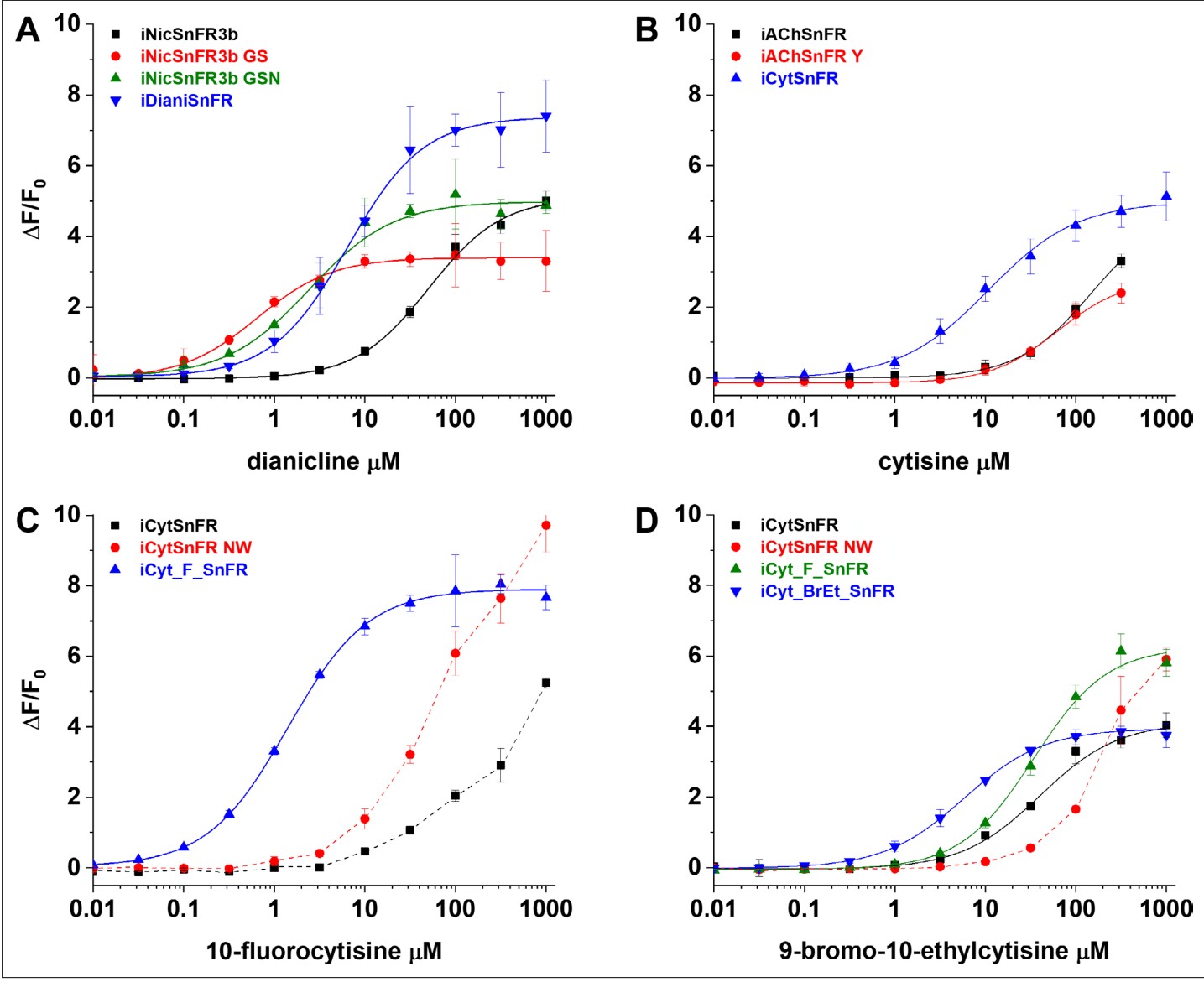

**Figure 2.** Nicotinic agonist intensity-based drug-sensing fluorescent reporter (iDrugSnFR) development. Dose–response relations on intermediate constructs using *E. coli* lysate were performed with respective drug partners to identify site-saturation mutagenesis (SSM) winners. (**A–D**) The progenitor biosensor is listed in black. Dashed lines indicate data that did not reach saturation at the concentrations tested; therefore, EC$_{50}$ and $\Delta F_{max}/F_0$ could not be determined. Development of (**A**) iDianiSnFR, (**B**) iCytSnFR, (**C**) iCyt_F_SnFR, and (**D**) iCyt_BrEt_SnFR.

cytisine were 0.61 ± 0.04 s$^{-1}$ (n = 36 areas, 105 cells) and 0.146 ± 0.006 s$^{-1}$ (n = 35 areas, n = 103 cells), respectively. The corresponding decay time constants ($\tau$ f$_{0ff}$, $\tau$ s$_{0ff}$) were 1.8 ± 0.1 s and 6.9 ± 0.2 s. Therefore, the temporal resolution of CytSnFR_PM sensor for cytisine was <10 s, adequate for the temporal resolution of the live-cell experiments presented below.

Interestingly, the decay waveform of the varenicline response was much slower than that for cytisine or ACh (**Figure 6A and D**). We pulsed 2 µM varenicline, >> the EC$_{50}$ of the steady-state response of the isolated protein (60 ± 10 nM) (**Figure 6D**). The values of the kf$_{off}$ and ks$_{off}$ were 0.9 ± 0.2 s$^{-1}$ and 0.0065 ± 0.0002 s$^{-1}$, respectively (n = 4 areas [nine cells]). The slower component dominated the decay phase, with a fractional amplitude of 85% ± 1%. Thus, the temporal resolution of the iCytSnFR_PM sensor for varenicline was in the minute range. In the live-cell experiments described below, it would not be possible to resolve the differences between varenicline at the PM and in the ER. The relatively high affinity of iCytisineSnFR for varenicline, which presumably arises in part from the increased lifetime of the varenicline-iDrugSnFR complex, has drawbacks. The temporal resolution

**Table 2.** Intensity-based drug-sensing fluorescent reporter (iDrugSnFR) dose–response relations versus a selected panel of nicotinic agonists.

ND, not determined. *, ** $EC_{50}$ and $\Delta F_{max}/F_0$ could not be determined from the data (*Figure 3*). Therefore, the upper limit to the S-slope is estimated from the data at the foot of the dose–response relation.

| Drug name | iDianiSnFR | | | iCytSnFR | | | iCyt_F_SnFR | | | iCyt_BrEt_SnFR | | |
|---|---|---|---|---|---|---|---|---|---|---|---|---|
| | $\Delta F_{max}/F_0$ | $EC_{50}$ (µM) | S-slope | $\Delta F_{max}/F_0$ | $EC_{50}$ (µM) | S-slope | $\Delta F_{max}/F_0$ | $EC_{50}$ (µM) | S-slope | $\Delta F_{max}/F_0$ | $EC_{50}$ (µM) | S-slope |
| Choline | 2.0 ± 0.1 | 84 ± 20 | < 0.1 | 5.8 ± 0.2 | 240 ± 30 | < 0.1 | 2.6 ± 0.1 | 18 ± 1 | 0.1 | 2.6 ± 0.1 | 12 ± 1 | 0.2 |
| Acetylcholine | 7.4 ± 1.0 | 660 ± 80 | < 0.1 | 2.9 ± 0.1 | 35 ± 3 | < 0.1 | 4.4 ± 0.3 | 222 ± 50 | < 0.1 | 2.5 ± 0.2 | 73 ± 6 | <0.1 |
| Cytisine | - | - | <0.1* | 7.3 ± 0.4 | 11 ± 1 | 0.7 | 4.4 ± 0.1 | 2.6 ± 0.3 | 1.7 | 4.7 ± 0.1 | 3.5 ± 0.2 | 1.3 |
| Dianicline | 4.7 ± 0.2 | 15 ± 1 | 0.3 | 6.5 ± 0.4 | 34 ± 4 | 0.2 | 2.3 ± 0.3 | 43 ± 6 | < 0.1 | 4–6 | >100 | <0.1** |
| Nicotine | 2.2 ± 0.1 | 440 ± 100 | < 0.1 | 6.4 ± 0.2 | 14 ± 2 | 0.5 | 4.7 ± 0.1 | 3.8 ± 0.2 | 1.2 | 4.8 ± 0.1 | 5.5 ± 0.2 | 0.9 |
| Varenicline | 2.4 ± 2.0 | 1200 ± 500 | < 0.1 | 6.5 ± 0.1 | 0.06 ± 0.01 | 110 | 7.1 ± 0.2 | 0.09 ± 0.02 | 79 | 5.3 ± 0.1 | 0.06 ± 0.01 | 88 |
| 10-Fluorocytisine | ND | ND | ND | ND | ND | ND | 2.3 ± 0.1 | 1.6 ± 0.3 | 1.4 | 3.0 ± 0.1 | 4.7 ± 0.3 | 0.6 |
| 9-Bromo-10-ethylcytisine | ND | ND | ND | ND | ND | ND | 3.1 ± 0.1 | 31 ± 2 | 0.1 | 3.6 ± < 0.1 | 4.2 ± 0.2 | 0.9 |

of iNicSnFR3a and iNicSnFR3b, which bind varenicline ~100-fold less tightly, is appropriate for subcellular experiments (*Shivange et al., 2019*). The previous experiments showing ER entry of varenicline used iNicSnFR3a and iNicSnFR3b (*Shivange et al., 2019*). For additional microperfusion data and analyses, see *Appendix 2—figures 1–3*.

## Characterization of nicotinic iDrugSnFRS in HeLa cells and primary mouse hippocampal culture

We examined the subcellular pharmacokinetics of the nicotinic agonists in mammalian cell lines and primary mouse hippocampal neurons. The nicotinic iDrugSnFRs were targeted to the PM (iDrugSnFR_PM) or the ER (iDrugSnFR_ER) as previously described (*Bera et al., 2019*; *Shivange et al., 2019*). We then performed a dose–response experiment using wide-field fluorescence imaging with each iDrugSnFR and its drug partner, sampling a range of concentrations covering a log scale surrounding the $EC_{50}$ as determined for the purified protein (*Figures 7 and 8*, *Videos 3–6*). iDianiSnFR showed a robust response to dianicline at the PM and the ER in HeLa cells across a range of concentrations (3.125–100 µM), and the speed was nearly limited by solution exchanges; there was a clear return to baseline fluorescence upon washout on the order of seconds after each drug application. At 100 µM, the PM and ER have a $\Delta F/F_0$ of ~1.2, but at lower concentrations, the ER displayed 30–75% of the signal detected at the PM, which may indicate a difference in membrane crossing (*Figure 7A*). Imaging in primary mouse hippocampal neurons demonstrated a similar trend (*Figure 8A*).

Cytisine showed slower entry into and exit from the ER of HeLa cells. The iCytSnFR_PM construct detected cytisine at concentrations from 0.078 to 80 µM and demonstrated a return to baseline fluorescence upon washout on the order of seconds after each drug application, reaching a maximum $\Delta F/F_0$ of ~2 at concentrations above 5 µM (*Figure 7B*). In contrast to the _PM construct, the iCytSnFR_ER construct only detected cytisine with a $\Delta F/F_0$ above the buffer control in the range of concentrations from 1.25 to 80 µM with a $\Delta F/F_0$, which was 25–50% of the maximum $\Delta F/F_0$ detected at the PM. Additionally, in the range of detectable concentrations of cytisine, the washout of cytisine was much slower than solution changes (*Figure 7B*). The incomplete washout persists even after several minutes and corresponds with previous suggestions that cytisine has low membrane permeability, as evidenced by its low brain penetration (*Rollema et al., 2010*).

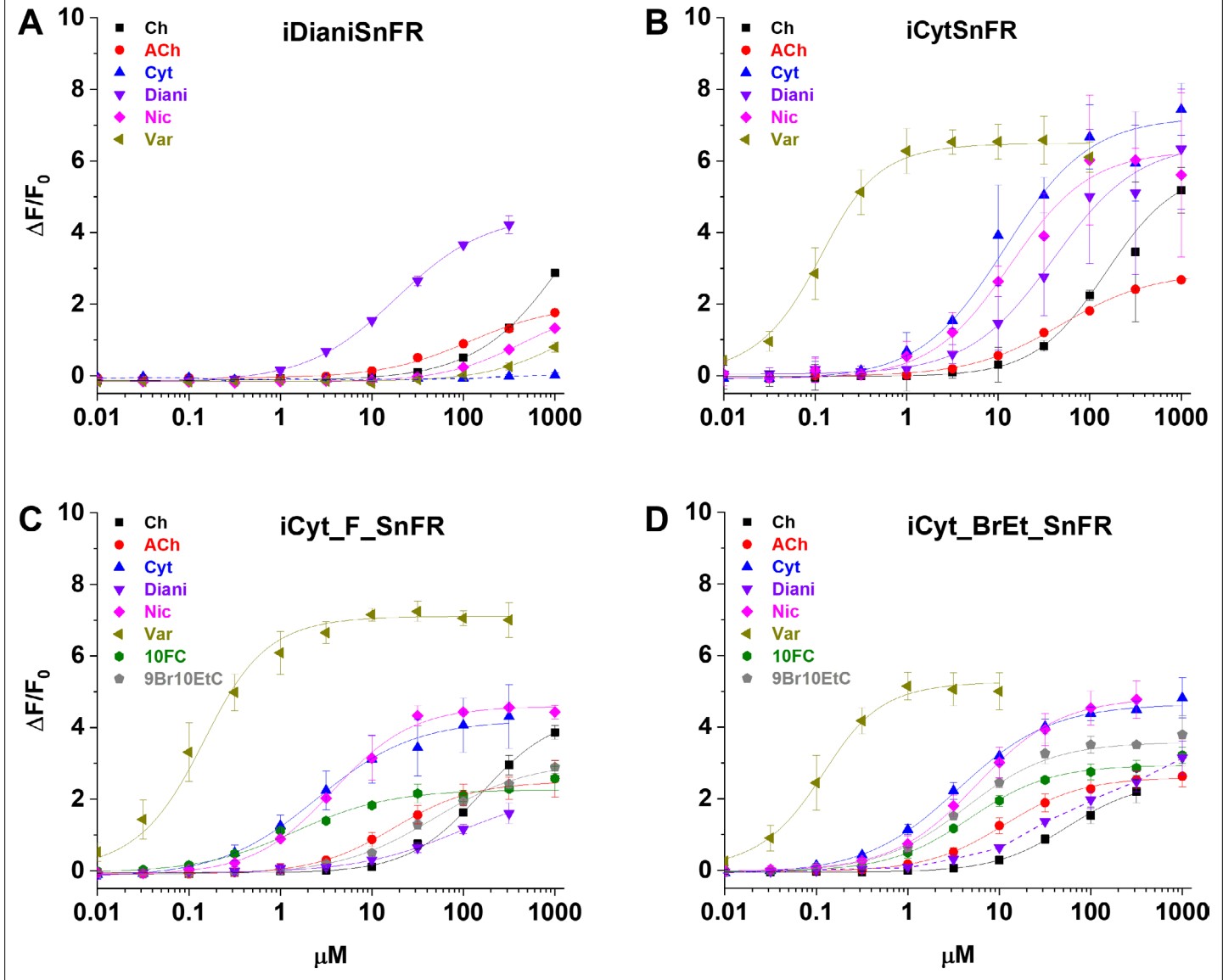

**Figure 3.** Dose–response relations of intensity-based drug-sensing fluorescent reporter (iDrugSnFR) protein versus a nicotinic agonist panel. (**A–D**) Relevant $EC_{50}$ values for each iDrugSnFR are listed in *Table 2*. Dashed lines indicate dose–response relations that did not approach saturation for the concentration ranges tested; therefore, $EC_{50}$ and $\Delta F_{max}/F_0$ could not be determined. (**A**) iDianiSnFR shows preference for dianicline, with some promiscuity for other nicotinic agonists. (**B**) iCytSnFR, (**C**) iCyt_F_SnFR, and (**D**) iCyt_BrEt_SnFR bind their drug partner, but also respond to other nicotinic agonists. Ch, choline; ACh, acetylcholine; Cyt, cytisine; Diani, dianicline; Nic, nicotine; Var, varenicline; 10FC, 10-fluorocytisine; 9Br10EtC, 9-bromo-10-ethylcytisine.

The online version of this article includes the following figure supplement(s) for figure 3:

**Figure supplement 1.** Dose–response relations of nicotinic agonist intensity-based drug-sensing fluorescent reporters (iDrugSnFRs) against select endogenous molecules.

In primary mouse hippocampal neurons, iCytSnFR detection of cytisine exhibited the same kinetic trends seen in HeLa cell experiments (*Figure 8B*). During cytisine application (60 s) from 0.078 to 80 µM, the iCytSnFR_PM fluorescence nearly reached a plateau, and during the washout (90–180 s), the fluorescence decayed back to baseline, though the decay slowed after removal of higher [cytisine]. The _PM construct reached a maximum $\Delta F/F_0$ of ~1.25 at 80 µM, which was approximately 60% of the signal observed in HeLa cell experiments (*Figure 8B*). The iCytSnFR_ER detection of cytisine in the ER reflected the trends seen in HeLa cells, with incomplete cytisine wash-in phases and prolonged cytisine washout phases. One observable difference was that the maximum $\Delta F/F_0$ (~1.25) of iCytSnFR_ER

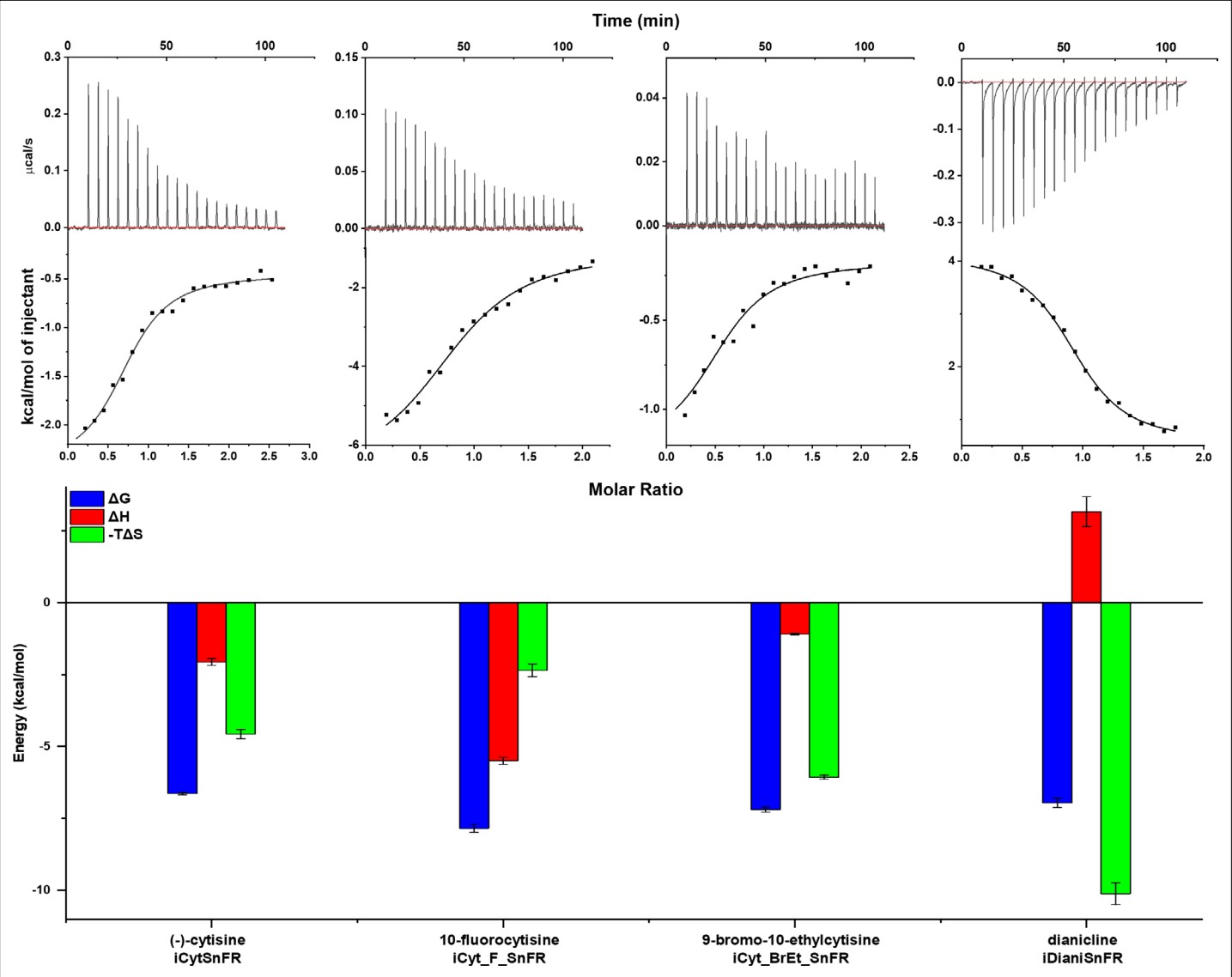

**Figure 4.** Isothermal titration calorimetry traces, fits, and thermodynamic data. Top row: exemplar heat traces of iCytSnFR, iCyt_F_SnFR, iCyt_BrEt_SnFR, and iDianiSnFR paired with their drug partners obtained by isothermal calorimetry. The heats for iCytSnFR, iCyt_F_SnFR, and iCyt_BrEt_SnFR were exothermic, while that for iDianiSnFR was endothermic. Middle row: the resulting fits for each iDrugSnFR:drug pair from the integrated heats comprising each series of injections. Bottom row: energy calculations. All iDrugSnFRs show exergonic reactions, but the relative enthalpic and entropic contributions vary among iDrugSnFRs. Data are from three separate runs, mean ± SEM. iDrugSnFR, intensity-based drug-sensing fluorescent reporter.

**Table 3.** Affinity, occupancy number, and thermodynamic data calculated from isothermal titration calorimetry.
Data are the mean ± SEM, three runs.

| Biosensor | KD (µM) | n | ΔH (kcal/mol) | -TΔS (kcal/mol) | ΔG (kcal/mol) |
|---|---|---|---|---|---|
| iCytSnFR | 13.7 ± 1.1 | 0.84 ± 0.05 | –2.1 ± 0.1 | –4.6 ± 0.2 | –6.6 ± 0.1 |
| iCyt_F_SnFR | 1.8 ± 0.5 | 0.83 ± 0.02 | –5.5 ± 0.1 | –2.4 ± 0.2 | –7.9 ± 0.1 |
| iCyt_BrEt_SnFR | 5.4 ± 0.8 | 0.69 ± 0.09 | –1.12 ± 0.03 | 6.1 ± 0.1 | –7.2 ± 0.1 |
| iDianiSnFR | 7.6 ± 1.4 | 0.92 ± 0.02 | 3.2 ± 0.5 | 10.1 ± 0.4 | –7.0 ± 0.2 |

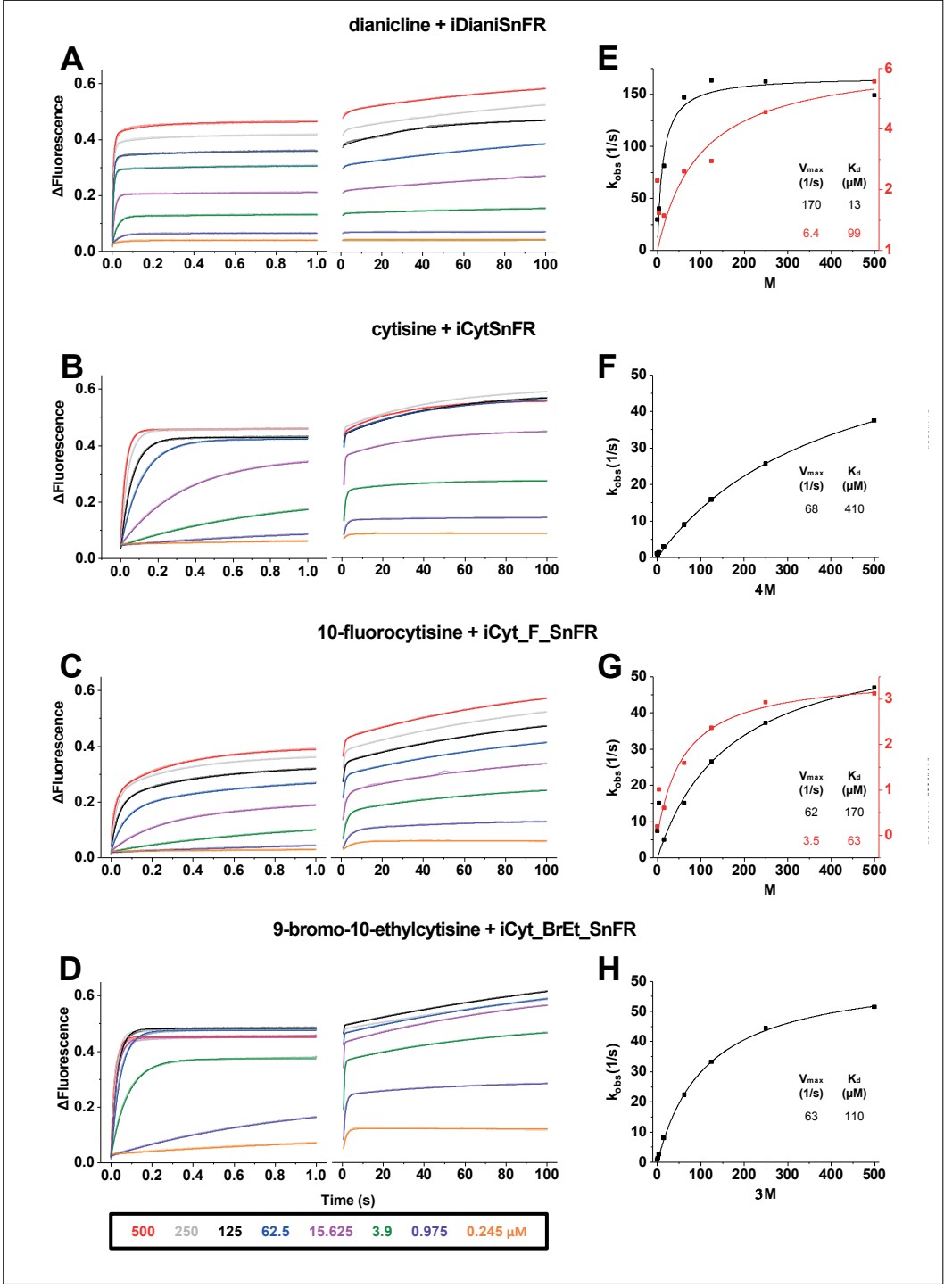

**Figure 5.** Stopped-flow fluorescence kinetic data for (**A**) iDianiSnFR, (**B**) iCytSnFR, (**C**) iCyt_F_SnFR, and (**D**) iCyt_BrEt_SnFR over 1 s and 100 s. Fluorescence was activated by mixing with the agonists, producing the indicated final concentrations. Stopped-flow data show a departure from first-order kinetics for this set of iDrugSnFRs. iDianiSnFR and iCyt_F_SnFR are fitted to a double exponential; iCytSnFR and iCyt_BrEt_SnFR are fitted to a single exponential. (**E–H**) Plots of the observed apparent rate constant $k_{obs}$ against [agonist] for the 1 s data obtained in (**A–D**).

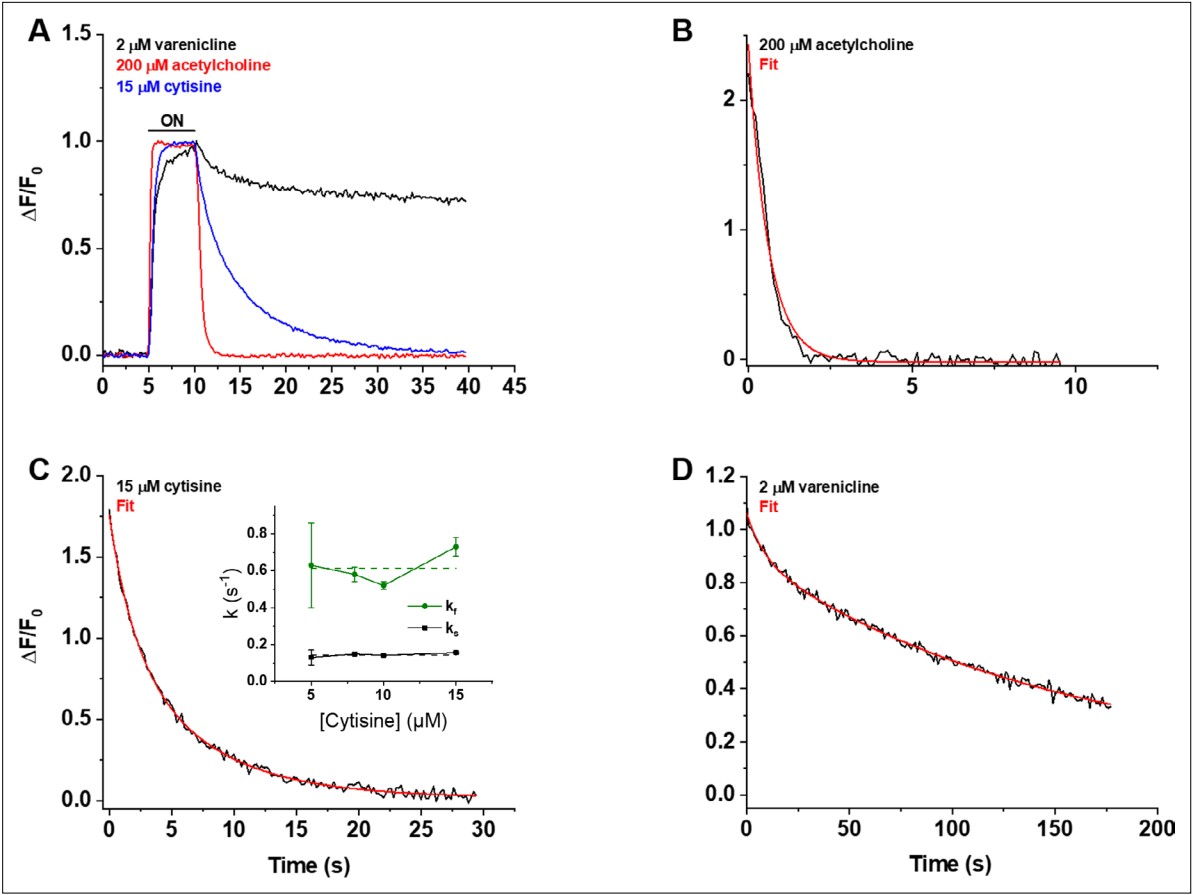

**Figure 6.** Decay of the iCytSnFR_PM responses after removal of ACh, cytisine, or varenicline. (**A**) The red, blue, and black traces are mean ΔF/F$_0$ values for the ACh (200 μM), cytisine (15 μM), and varenicline (2 μM) responses as a function of time (n = 4–10 areas per ligand). The ΔF/F$_0$ was normalized to the peak response for each ligand. Sampling rate was 5 frames/s. Ligand was applied for 5 s, denoted by the black horizontal bar above the traces. (**B–D**) Examples of the decay phase of the response to ACh (200 μM), cytisine (15 μM), and varenicline (2 μM) in individual areas (black traces in each panel). Red lines are fits to the sum of one or two negative exponential terms and a constant (red lines in each panel) using nonlinear least-squares regression. (**B**) The decay of the ACh (200 μM) response (n = 1 area, 3 cells) was monophasic with a single time constant ($\tau_{off}$) of 0.61 ± 0.02 s (± SE, n = 86 frames, sampling rate of 9.8 frames/s). The red line is a fit to the sum of a negative exponential component (R$^2$ of 0.98). (**C**) The decay of the cytisine (15 μM) response (n = 1 area, four cells) was biphasic with time constants ($\tau$ f$_{off}$, $\tau$ s$_{off}$) of 1.9 ± 0.2 and 6.6 ± 0.5 s (n = 149 frames, sampling rate of 5 frames/s). The red line is a fit to the sum of two negative exponential components and a constant (R$^2$ of 0.996). It was significantly better than that of the sum of a single negative exponential term and a constant (F-test, p<0.05). The relative amplitude of the slower decay component (A$_s$/(A$_s$+ A$_f$), where A$_s$ is amplitude of the slower component of decay in units of ΔF/F$_0$ and A$_f$ is amplitude of the faster component) was 61%. Inset: neither rate constant changed significantly over the [cytisine] range from 5 to 15 μM. Dashed lines give the average over this range. (**D**) The decay of the varenicline (2 μM) response (n = 1 area, three cells) was also biphasic with a $\tau$ f$_{off}$ and $\tau$ s$_{off}$ of 9 ± 1 s and 150 ± 10 s (n = 178 frames, sampling rate of 1 frame/s), respectively. The A$_s$/(A$_s$+ A$_f$) was 83%. The red line is a fit to the sum of two negative exponential terms and a constant (R$^2$ of 0.994), and it was significantly better than that to the sum of a single negative exponential term and a constant (F-test, p<0.05).

reached a similar maximum to that of iCytSnFR_PM in neurons, which was not observed in HeLa cell experiments (*Figure 7B*).

In preliminary HeLa cell experiments with varenicline applied to iCytSnFR, we found much slower kinetics that differed little between the _ER and _PM constructs (data not shown). These findings, which vitiated the use of the iCytSnFR-varencline pair in the cellular experiments, are consistent with the markedly slow kinetics of varenicline-iCytSnFR interactions in the microperfusion experiments (see above).

iCyt_F_SnFR targeted to the PM and ER showed characteristics similar to iCytSnFR in HeLa cells. The _PM construct detected 10-fluorocytisine across a range of concentrations with a return to baseline fluorescence between applications, while the _ER construct detected 10-fluorocytisine with ΔF/F$_0$ values that were only 25–33% of those detected at the PM (*Figure 7C*). Similar to the iCytSnFR_ER detection of cytisine, the iCyt_F_SnFR_ER detection of 10-fluorocytisine was much slower than

solution changes and did not return to baseline between applications, though the washout occurs on the order of minutes, rather than tens of minutes as with iCytSnFR_ER (*Figure 7C*). The difference in PM and ER detection of 10-fluorocytisine again shows decreased membrane permeability into HeLa cells compared to other drugs we have examined with other iDrugSnFRs. Overall, the detection of 10-fluorocytisine with iCyt_F_SnFR in primary hippocampal culture resembled our data with iCyt_F_SnFR in HeLa cells. Nevertheless, there were distinct differences (*Figure 8C*), such as a decreased maximum $\Delta F/F_0$ in the iCyt_F_SnFR_PM construct and a similar maximum $\Delta F/F_0$ of ~1 for both the _ER and _PM constructs. Additionally, the decay of the iCyt_F_SnFR responses lasted tens of minutes, resembling the iCytSnFR_ER data in primary hippocampal culture.

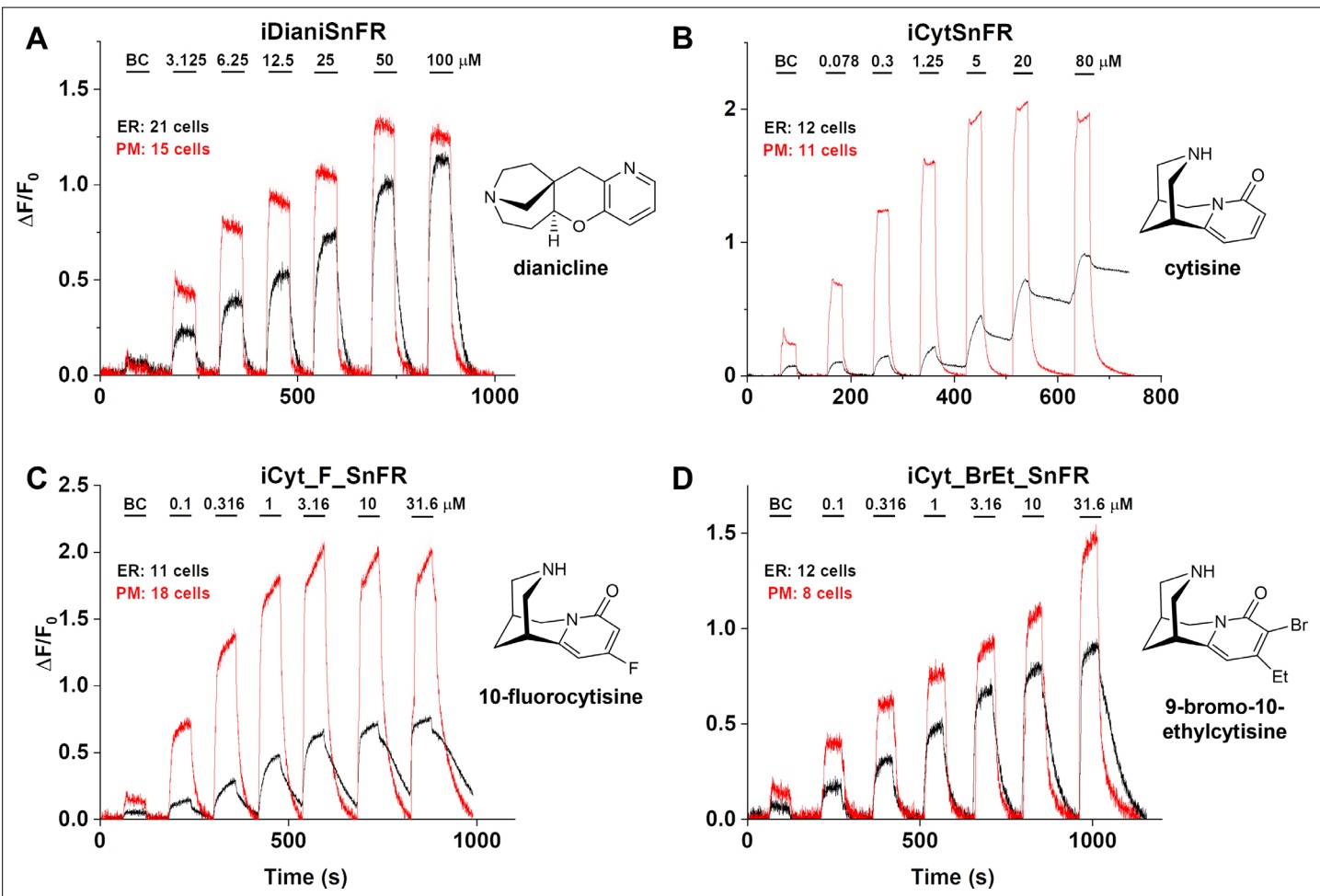

**Figure 7.** Nicotinic agonist intensity-based drug-sensing fluorescent reporter (iDrugSnFR) dose–response relations in HeLa cells. (**A–D**) Each iDrugSnFR detects its drug partner at the plasma membrane (PM) and endoplasmic reticulum (ER) of HeLa cells at the concentrations sampled. BC, buffer control. SEM of data are indicated by semi-transparent shrouds around traces where trace width is exceeded. (**A**) iDianiSnFR detects dianicline with a return to baseline fluorescence between drug applications. (**B**) iCytSnFR detection at the PM returns to baseline fluorescence between applications, while detection at the ER shows incomplete wash-in and washout. (**C**) iCyt_F_SnFR fluorescence response to the presence of 10-fluorocytisine in the ER also shows an incomplete washout between applications. (**D**) iCyt_BrEt_SnFR detects 9-bromo-10-ethylcytisine with wash-in and washout fluorescence similar to the pattern seen in iDianiSnFR.

The online version of this article includes the following figure supplement(s) for figure 7:

**Figure supplement 1.** Traces of fluorescence responses during time-resolved low-concentration dose–response relations for nicotinic agonists in HeLa cells.

**Figure supplement 2.** Dose–response relations for iCytSnFR and iCyt_F_SnFR against nicotine in HeLa cells.

**Figure supplement 3.** Spinning disk laser scanning confocal inverted microscope images of nicotinic agonist intensity-based drug-sensing fluorescent reporters (iDrugSnFRs) in HeLa cells.

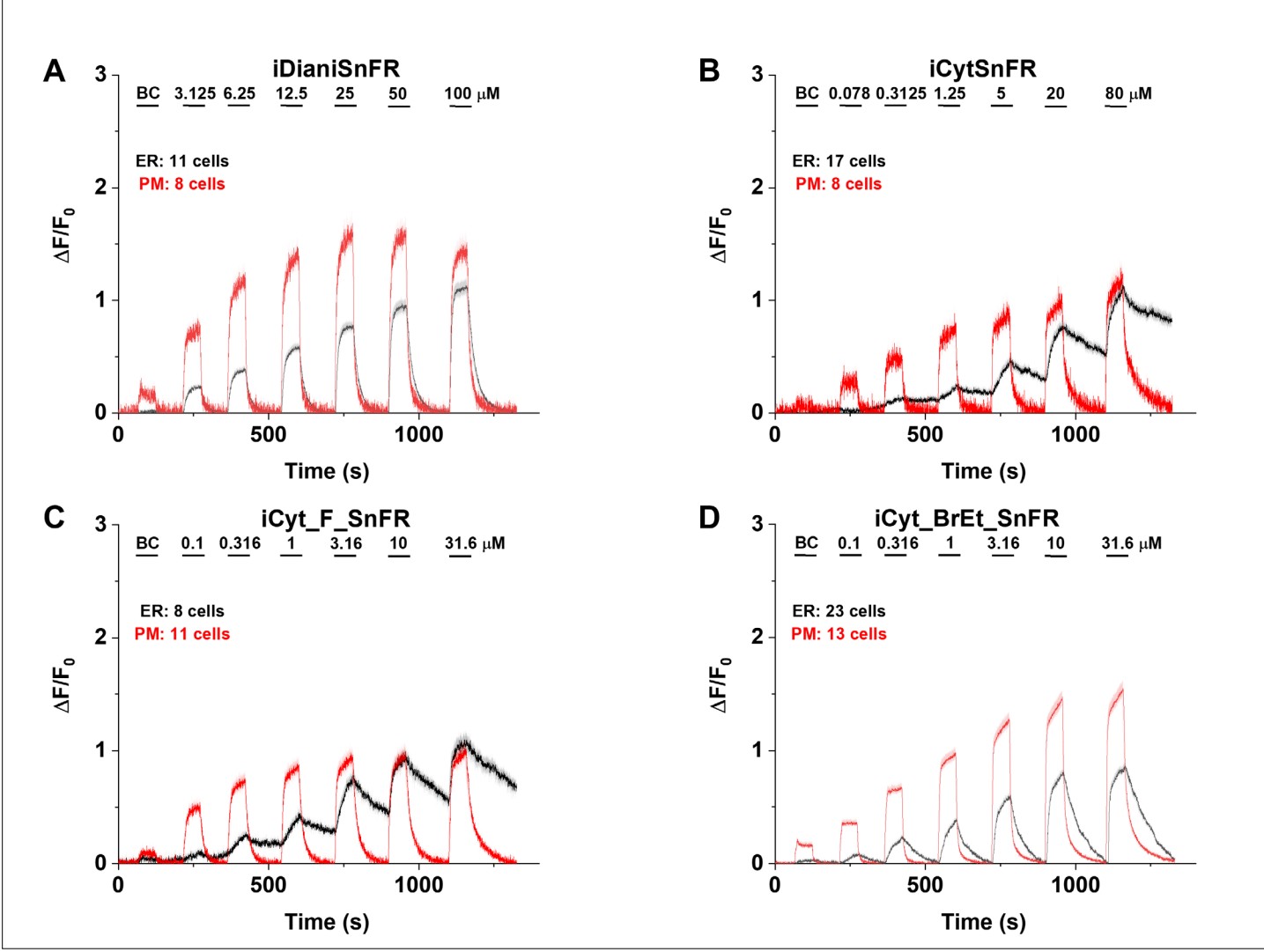

**Figure 8.** Nicotinic agonist intensity-based drug-sensing fluorescent reporter (iDrugSnFR) dose–response experiments in mouse primary hippocampal neurons transduced with AAV9-hSyn iDrugSnFR. Cultured primary mouse hippocampal neurons were transduced with endoplasmic reticulum (ER)- or plasma membrane (PM)-targeted constructs. BC, buffer control. SEM of data are indicated by semi-transparent shrouds around traces where trace width is exceeded. (**A–D**) Each iDrugSnFR detects its drug partner at the PM and ER over the concentrations sampled. (**A**) iDianiSnFR detects dianicline with a return to baseline fluorescence between drug applications. (**B**) iCytSnFR detection at the PM returns to baseline fluorescence between applications, while detection at the ER shows an incomplete washout. (**C**) iCyt_F_SnFR fluorescence response to the presence of 10-fluorocytisine in the ER also shows an incomplete washout between applications. (**D**) iCyt_BrEt_SnFR_ER detects 9-bromo-10-ethylcytisine with a wash-in and decay intermediate between iDianiSnFR and the other two cytisine derivatives.

The online version of this article includes the following figure supplement(s) for figure 8:

**Figure supplement 1.** Spinning disk laser scanning confocal inverted microscope images of nicotinic agonist intensity-based drug-sensing fluorescent reporters (iDrugSnFRs) in primary mouse hippocampal neurons.

9-Bromo-10-ethylcytisine showed a kinetic profile resembling dianicline. iCyt_BrEt_SnFR_PM responses to 9-bromo-10-ethylcytisine (0.1–31.6 µM) were nearly limited by solution exchanges with a return to baseline fluorescence on the order of seconds, and a maximum $\Delta F/F_0$ of ~1.5 at 31.6 µM. iCyt_BrEt_SnFR_ER also detected 9-bromo-10-ethylcytisine over this range of concentrations and returned to baseline fluorescence between applications (*Figure 7D*). $\Delta F/F_0$ values for iCyt_BrEt_SnFR_ER were 50–75% of the $\Delta F/F_0$ values detected by iCyt_BrEt_SnFR_PM, which indicated that 9-bromo-10-ethylcytisine crossed into and out of cells readily (*Figure 7D*). Imaging in primary mouse hippocampal neurons revealed the same trend (*Figure 8D*).

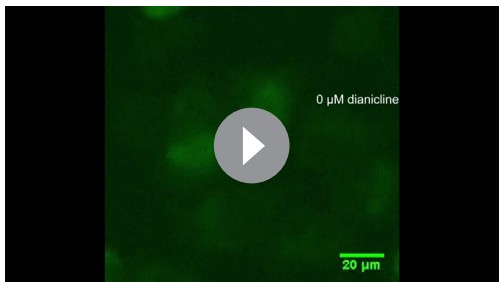

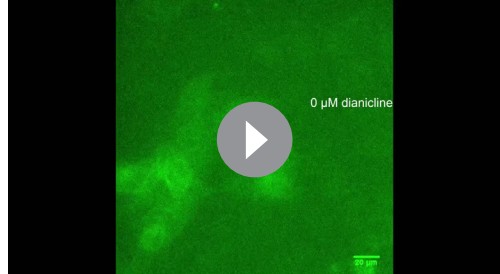

**Video 3.** iDianiSnFR_ER dose-response relations in HeLa cells. The dianicline concentrations are shown. The scale bar is shown. The video is 25-fold faster than real time.

https://elifesciences.org/articles/74648/figures#video3

**Video 4.** iDianiSnFR_PM dose-response relations in HeLa cells. The dianicline concentrations are shown. The scale bar is shown.The video is 25-fold faster than real time.

https://elifesciences.org/articles/74648/figures#video4

To more fully examine the membrane-crossing properties of the nicotinic agonists, we recorded the fluorescence waveforms for several drugs at concentrations between 0.1 and 3.16 µM with much longer application times and washout times than in the above experiments (*Figure 7—figure supplement 1*). With these conditions, the fluorescence signals suggested complete washout of each nicotinic agonist from the ER of HeLa cells. However, it is noteworthy that even when applied at concentrations as low as 0.1 µM and 0.316 µM, cytisine and 10-fluorocytisine require washout times of several minutes from the ER. In contrast, the iDrugSnFR localized to the PM shows a rapid return to baseline after drug application.

Because the data of *Figures 7B and 8B* indicated that iCytSnFR_PM functions as expected from stopped-flow and millisecond perfusion, we applied additional experiments to ensure that our observations of drug entry and exit from the ER were not the result of idiosyncratic biosensor function or folding in the ER. iCytSnFR and iCyt_F_SnFR both bind nicotine in the same concentration range as cytisine (though with lower $\Delta F/F_0$). After transfection of _PM and _ER constructs for each sensor into HeLa cells, we performed time-resolved imaging for pulses of 0.1–31.6 µM nicotine (*Figure 7—figure supplement 2*). These nicotine waveforms resembled those already published with iNicSnFR3a and iNicSnFR3b (*Shivange et al., 2019*), confirming that iCytSnFR_ER functions as expected with a more permeant nicotinic drug. Thus, the slower kinetics for iCytSnFR_ER with cytisine and iCyt_F_SnFR_ER with 10-fluorocytisine arise because these drugs cross membranes more slowly.

To examine localization of the _PM and _ER constructs at higher optical resolution, we imaged HeLa cells and primary mouse hippocampal culture using a spinning disk laser scanning inverted confocal microscope. As previously observed, ER-targeted iDrugSnFR was retained in the ER (*Figure 7—figure supplement 3*, *Figure 8—figure supplement 1*; *Shivange et al., 2019*). The iDrugSnFR constructs targeted to the PM showed correct localization, with some iDrugSnFR observed in the cell interior

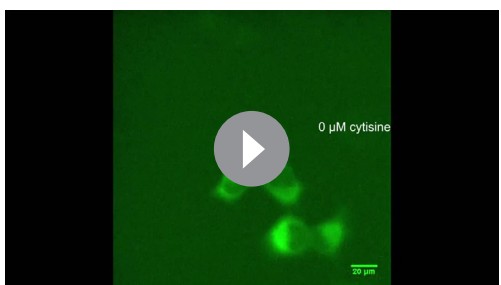

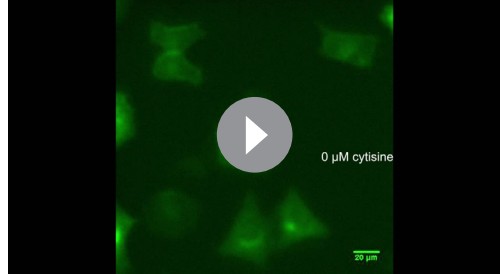

**Video 5.** iCytSnFR_ER dose-response relations in HeLa cells. The cytisine concentrations are shown. The scale bar is shown. The video is 25-fold faster than real time.

https://elifesciences.org/articles/74648/figures#video5

**Video 6.** iCytSnFR_PM dose-response relations in HeLa cells. The cytisine concentrations are shown. The scale bar is shown. The video is 25-fold faster than real time.

https://elifesciences.org/articles/74648/figures#video6

(most likely as part of the cellular membrane trafficking system; *Figure 7—figure supplement 3*, *Figure 8—figure supplement 1*).

Several complexities in the HeLa cell and neuron experiments imposed uncertainties on our kinetic analyses. These complexities include the limitations of solution changes, diffusion within cytoplasm, unknown mixing at the surface facing the coverslip, and corrections for baseline drift due to bleaching. We restrict the quantitative comparisons to the estimate that cytisine and 10-fluorocytisine cross the membrane >30-fold more slowly than the other drugs tested.

## Discussion

### Membrane permeation of molecules with low $logD_{pH7.4}$

The experiments show, to our knowledge, the first time-resolved measurements of membrane permeation for drugs in the $logD_{pH7.4}$ range less than –1. Most orally available drugs have $logD_{pH7.4}$ values between 2 and 4 (*Smith et al., 2012*). Cytisine, varenicline, dianicline, and the cytisine analogs studied here have calculated membrane partition coefficients some 3–6 orders of magnitude lower. These values and their order vary according to the algorithm, partially because of uncertainties in predicting $pK_a$ (*Pieńko et al., 2016*); here, we provide values calculated by Chemicalize (see Materials and methods): 10-fluorocytisine, –2.70; cytisine, –2.64; dianicline, –1.29; varenicline, –1.27; 9-bromo-10-ethylcytisine, –1.13. It is remarkable that drugs with such low calculated partition coefficients do cross membranes on a time scale of seconds (9-bromo-10-ethylcytisine, varenicline, dianicline) to minutes (10-fluorocytisine, cytisine). According to some (but not all) algorithms, the calculated $logD_{pH7.4}$ values fall in the same two classes as the measured kinetics of membrane permeability: 10-fluorocytisine and cytisine are the slowest, and only these two agonists have $logD_{pH7.4}$ values < –2. These observations support previous work suggesting that differences among chemical properties of nicotinic partial agonists correlate with drug permeation into the cerebrospinal fluid (CSF) after peripheral administration in mice (*Rollema et al., 2010*).

### The iDrugSnFR paradigm

The iDrugSnFRs are sensitive enough to allow experiments near the experimentally determined (or otherwise projected) concentration in the human blood and CSF (*Astroug et al., 2010*; *Jeong et al., 2018*; *Rollema et al., 2010*). The iDrugSnFRs have the advantage that they measure free aqueous ligand concentration ('activity'), as sensed by nAChRs. Targeting sequences provide for visualization within the lumen of organelles – here, the ER.

The experiments do not use radiolabeled drugs, in vivo microdialysis or other experiments on live animals, or mass spectrometry-liquid chromatography instruments. Once protein design has given an entry into a class of iDrugSnFRs, straightforward optimization at the binding site produces the desired, selective iDrugSnFRs for individual molecules. For drugs that bind at orthosteric cholinergic sites (both nicotinic and muscarinic), we anticipate that a collection of tens, rather than hundreds, of iDrugSnFRs will suffice to detect all present and future ligands. The experiments use standard, modest-power fluorescence microscopes. Cultured cell lines yield data comparable to cultured neurons.

We comment on varenicline. None of the biosensors in *Table 1* were evolved to bind varenicline; yet it binds to some iDrugSnFRs with nanomolar $EC_{50}$. Only iDianiSnFR, which lacks His68, binds varenicline with $EC_{50}$ > 10 μM. Even tighter binding has been achieved with varenicline derivatives at mutated ligand-gated channels (*Magnus et al., 2019*). On the one hand, the cellular experiments described here cannot use iDrugSnFR pairs with dissociation rate constants less than ~0.1 $s^{-1}$, corresponding to an $EC_{50}$ of less than ~100 nM. On the other hand, all known neural drugs leave the human body and brain much more slowly, with rates determined primarily by metabolism; even 'fast' nicotine metabolizers display time constants of ~1200 s (*Dempsey et al., 2004*). Highly sensitive, tightly binding, reagentless iDrugSnFRs will be used in studies on personal pharmacokinetics in biofluids.

### Structure–function relations for nicotinic and other iDrugSnFRs

This study shows that the amine group of nicotinic ligands makes equidistant cation-π interactions with two tyrosine residues (Tyr65, Tyr357), and this is confirmed by higher-resolution (1.5–1.7 Å) structures of varenicline, acetylcholine, and choline crystallized with isolated PBP moieties (PDB 7S7X, SV1R, 7S7Z, respectively; see also 3R6U, 6EYQ, and 3PPQ). Cation-π interactions also occur for cholinergic

and/or nicotinic ligands in nAChRs (*Morales-Perez et al., 2016*; *Post et al., 2017*), the acetylcholine-binding protein (*Celie et al., 2004*), PBPs (*Schiefner et al., 2004*), and muscarinic receptors (*Haga et al., 2012*). We also observe that the protonated amine of varenicline makes a hydrogen bond to a backbone carbonyl group, another similar theme in acetylcholine binding protein (*Celie et al., 2004*) and nAChRs (*Xiu et al., 2009*).

This study presents a general step forward in understanding the structure–function relations of iDrugSnFRs. The chromophore in the cpGFP moiety of most present iDrugSnFRs (this paper, iGluSnFR, iSeroSnFR) contains a tyrosine in an extended π system (*Ormö et al., 1996*; *Tsien, 1998*). The photophysics of the chromophore depends strongly on the surrounding water molecules and side chains (*Brejc et al., 1997*; *Tsien, 1998*). We found that Glu78 in Linker 1 changes its orientation: in the liganded state, it interacts with two positively charged residues (Lys97 and Arg99) on the surface of the cpGFP; and in the apo state, Glu78 has moved ~14 Å to form a hydrogen bonding interaction with the tyrosine moiety of the chromophore. Presumably the liganded state of iNicSnFR3adt allows for a water molecule to hydrogen bond with the hydroxy group of the chromophore, promoting its fluorescence; but this water molecule is replaced by protonated Glu78 in the unliganded state, which leads to nonfluorescent state of cpGFP, as suggested by *Nasu et al., 2021*.

While we cannot resolve the protonation-deprotonation event, the available functional data show good support for its occurrence, as follows. (1) The apo form of the iDrugSnFR increases its $F_0$ by 10-fold per pH unit (*Shivange et al., 2019*), as though when deprotonated, Glu78 leaves the 'candle snuffer' position and moves to make the salt bridges with Lys97 and Arg99. (2) The $EC_{50}$ for the ligand decreases by 10-fold per pH unit (*Shivange et al., 2019*), as though the conformation of the linker that forms the salt-bridge form is also the closed, liganded, fluorescent form of the PBP. Other observations favor the crucial role of the Glu78–chromophore interaction. (3) Only glutamate functions in position 78 of iSeroSnFR (*Unger et al., 2020*). (4) The mTurquoise variant in iGluSnFR, which has a tryptophan chromophore, requires entirely different linkers (*Marvin et al., 2018*).

## Challenges at the intersection of pharmaceutical science and nicotine addiction science

Our measurements show that nicotinic ligands with $logD_{pH7.4} < \sim{-}2$ cross membranes much more slowly than do ligands with $logD_{pH7.4} > \sim{-}2$. These measurements have two, possibly opposing, implications for future smoking-cessation drugs. On the one hand, α4β2 agonists that enter the ER, like nicotine and varenicline, upregulate nAChRs (*Turner et al., 2011*), which may be necessary and sufficient for addiction (*Henderson and Lester, 2015*); and maintenance of upregulation by varenicline may help to explain its suboptimal quit rate. On the other hand, ligands that do not enter the ER are also unlikely to enter the brain and therefore unlikely to be useful for smoking cessation.

Smoking-cessation drugs must also contend with other ER-based processes. (1) Most drug metabolism takes place in the ER and (2) upregulation occurs at a sustained agonist concentration in the ER some hundredfold lower than the extracellular concentrations that transiently activate α4β2 nAChRs (*Kuryatov et al., 2005*).

Given these challenges, further progress may be possible now that we have two types of real-time, living cellular preparations. (1) For decades, cellular preparations have been available to measure nAChR pharmacodynamics and upregulation. (2) Now, the iDrugSnFRs present a paradigm to measure cellular and subcellular pharmacokinetics. The iDrugSnFR paradigm will be useful beyond the explicit case of nicotine addiction, with application to other exogenous neural drugs.

# Materials and methods

## Key resources table

| Reagent type (species) or resource | Designation | Source or reference | Identifiers | Additional information |
|---|---|---|---|---|
| Strain, strain background (*Escherichia coli*) | BL21(DE3) | Agilent Technologies, Santa Clara, CA | 200131 | Chemically competent |
| Cell line (*Homo sapiens*) | HeLa | ATCC | CCL-2; RRID:CVCL_0030 | |
| Cell line (*H. sapiens*) | HEK293T | ATCC | CRL-3216; RRID:CVCL_0063 | |

*Continued on next page*

*Continued*

| Reagent type (species) or resource | Designation | Source or reference | Identifiers | Additional information |
|---|---|---|---|---|
| Biological sample (*Mus musculus*) | Primary hippocampal neurons | Caltech animal facilities | RRID:IMSR_JAX:000664 | Freshly isolated from *Mus musculus* |
| Recombinant DNA reagent | iAChSnFR | Loren Looger | Addgene: 137955 | Obtainable through Addgene |
| Recombinant DNA reagent | iDianiSnFR | This paper | Addgene: 177741 | Obtainable through Addgene |
| Recombinant DNA reagent | iCytSnFR | This paper | Addgene: 177738 | Obtainable through Addgene |
| Recombinant DNA reagent | iCyt_F_SnFR | This paper | Addgene: 177739 | Obtainable through Addgene |
| Recombinant DNA reagent | iCyt_BrEt_SnFR | This paper | Addgene: 177740 | Obtainable through Addgene |
| Recombinant DNA reagent | pCMV(MinDis)-iDianiSnFR_PM | This paper | Addgene: 177751 | Obtainable through Addgene |
| Recombinant DNA reagent | pCMV(MinDis)-iDianiSnFR_ER | This paper | Addgene: 177750 | Obtainable through Addgene |
| Recombinant DNA reagent | pCMV(MinDis)-iCytSnFR_PM | This paper | Addgene: 177743 | Obtainable through Addgene |
| Recombinant DNA reagent | pCMV(MinDis)-iCytSnFR_ER | This paper | Addgene: 177742 | Obtainable through Addgene |
| Recombinant DNA reagent | pCMV(MinDis)-iCyt_F_SnFR_PM | This paper | Addgene: 177745 | Obtainable through Addgene |
| Recombinant DNA reagent | pCMV(MinDis)-iCyt_F_SnFR_ER | This paper | Addgene: 177744 | Obtainable through Addgene |
| Recombinant DNA reagent | pCMV(MinDis)-iCyt_BrEt_SnFR_PM | This paper | Addgene: 177747 | Obtainable through Addgene |
| Recombinant DNA reagent | pCMV(MinDis)-iCyt_BrEt_SnFR_ER | This paper | Addgene: 177746 | Obtainable through Addgene |
| Recombinant DNA reagent | pAAV9-hSyn-iDianiSnFR_PM | This paper | Addgene: 177759 | Obtainable through Addgene |
| Recombinant DNA reagent | pAAV9-hSyn-iDianiSnFR_ER | This paper | Addgene: 177758 | Obtainable through Addgene |
| Recombinant DNA reagent | pAAV9-hSyn-iCytSnFR_PM | This paper | Addgene: 177753 | Obtainable through Addgene |
| Recombinant DNA reagent | pAAV9-hSyn-iCytSnFR_ER | This paper | Addgene: 177752 | Obtainable through Addgene |
| Recombinant DNA reagent | pAAV9-hSyn-iCyt_F_SnFR_PM | This paper | Addgene: 177755 | Obtainable through Addgene |
| Recombinant DNA reagent | pAAV9-hSyn-iCyt_F_SnFR_ER | This paper | Addgene: 177754 | Obtainable through Addgene |
| Recombinant DNA reagent | pAAV9-hSyn-iCyt_BrEt_SnFR_PM | This paper | Addgene: 177757 | Obtainable through Addgene |
| Recombinant DNA reagent | pAAV9-hSyn-iCyt_BrEt_SnFR_ER | This paper | Addgene: 177756 | Obtainable through Addgene |
| Commercial assay or kit | Phusion High-Fidelity PCR Kit | New England Biolabs | E0553L | |
| Commercial assay or kit | Q5 Site-Directed Mutagenesis Kit | New England Biolabs | E0554S | |
| Commercial assay or kit | QIAprep Spin Miniprep Kit | QIAGEN SCR_008539 | 27104 | |
| Commercial assay or kit | EndoFree Plasmid Maxi Kit | QIAGEN SCR_008539 | 12362 | |
| Commercial assay or kit | QIAquick PCR Purification Kit | QIAGEN SCR_008539 | 28104 | |
| Commercial assay or kit | QIAquick Gel Extraction Kit | QIAGEN SCR_008539 | 28704 | |
| Commercial assay or kit | AAVpro Purification Kit | Takara Bio Inc. | 6666 | |
| Commercial assay or kit | PACT premier | Molecular Dimensions | MD1-29 | |
| Chemical compound, drug | 10-Fluorocytisine | Tim Gallagher | | *Rego Campello et al., 2018* |
| Chemical compound, drug | 9-Bromo-10-ethylcytisine | Tim Gallagher | | *Rego Campello et al., 2018* |
| Chemical compound, drug | Lipofectamine 2000 Transfection Reagent | Thermo Fisher Scientific | 11668027 | |
| Chemical compound, drug | Lipofectamine 3000 Transfection Reagent | Thermo Fisher Scientific | L3000015 | |
| Software, algorithm | Spartan'20 | Wavefunction, Inc. | RRID:SCR_014901 | |
| Software, algorithm | NanoAnalyze | TA Instruments | | https://www.tainstruments.com/sw/nano_analyze.html |
| Software, algorithm | OriginPro 2018 | OriginLab | RRID:SCR_014212 | |
| Software, algorithm | KaleidaGraph | Synergy | RRID:SCR_014980 | |
| Software, algorithm | ImageJ | NIH | RRID:SCR_003070 | |
| Software, algorithm | XDS Program Package | MPI for Medical Research, Heidelberg | RRID:SCR_015652 | |
| Software, algorithm | Phenix | Phenix | RRID: SCR_014224, SCR_016736 | *Adams et al., 2010* |

| Reagent type (species) or resource | Designation | Source or reference | Identifiers | Additional information |
|---|---|---|---|---|
| Software, algorithm | Coot | MRC Laboratory of Molecular Biology | RRID:SCR_014222 | *Emsley et al., 2010* |

## Crystallography

The gene encoding the full-length biosensor iNicSnFR3a was previously cloned into a bacterial expression vector (*Shivange et al., 2019*). To improve crystallization, we deleted the N-terminal HA tag and the N-terminal Myc tag, forming the constructs with the suffix 'dt.' These deletions were carried out with the Q5 Site-Directed Mutagenesis Kit (New England Biolabs, Ipswich, MA). All proteins were overexpressed in *E. coli* BL21-gold (DE3) cells (Agilent Technologies, Santa Clara, CA) using ZYM-5052 autoinduction media (*Studier, 2005*). Cells were collected by centrifugation and stored at –80°C until use.

For purification, frozen cell pellets were resuspended in lysis buffer containing 100 mM NaCl, 20 mM Tris, pH 7.5, 20 mM imidazole, pH 7.5, 5 mM β-mercaptoethanol (BME), lysozyme, DNase, and protease inhibitor tablet. The resuspended cells were lysed by freezing and thawing using liquid nitrogen and a room temperature water bath for three cycles. Intact cells and cell debris were removed by centrifugation at ~20,000 × $g$ for 40 min at 4°C. The supernatant was collected and loaded onto a prewashed Ni NTA column with wash buffer at 4°C. Ni NTA wash buffer contained 100 mM NaCl, 20 mM Tris, pH 7.5, 30 mM imidazole, pH 7.5, and 5 mM BME. Elution was achieved using the same buffer with 300 mM imidazole, pH 7.5. The eluted sample was further purified by size-exclusion chromatography using HiLoad 16/60 Superdex 200 in the same buffer without imidazole and BME. Peak fractions were collected and concentrated to ~50 mg/ml with Amicon Ultra 15 filter unit (Millipore, Burlington, MA) with 10 kDa cutoff.

For all constructs, initial crystallization screening was carried out with 40 mg/ml protein in the presence and absence of 10 mM nicotine or varenicline. iNicSnFR3adt crystallized separately with 10 mM nicotine and varenicline in PACT premier (Molecular Dimensions, Sheffield, England), condition #96 with 0.2 M sodium malonate dibasic monohydrate, 0.1 M Bis-Tris Propane, pH 8.5, and 20% polyethylene glycol (PEG) 3350 at 20°C. Crystals of iNicSnFR3adt grew within 2 weeks of crystallization in a hexagonal rod shape with dimensions of ~80 μm × 80 μm × 300 μm. Crystals were harvested and cryo-protected in 25% ethylene glycol, 0.2 M sodium malonate dibasic monohydrate, 0.1 M Bis-Tris Propane pH 8.5, and 20% PEG 3350. Phase information was obtained through soaking with KI before cryo-protection. The unliganded iNicSnFR3adt crystallized in Morpheus (Molecular Dimensions), condition #92 with 2.5% PEG 1000, 12.5% PEG 3350, 12.5% 2-methyl-2,4-pentanediol, 0.02 M of each amino acid, and 0.1 M MOPS/HEPES-Na, pH 7.5 at 23°C with no further optimization.

X-ray datasets were collected at Stanford Synchrotron Radiation Laboratory beamline 12-2 and Advanced Light Source beamline 5.0.2 using Pilatus 6M detectors. All datasets were processed and integrated with XDS (*Kabsch, 2010*) and scaled with Aimless (*Winn et al., 2011*). For iNicSnFR3adt, molecular replacement was carried out using domains of the unliganded structure (PDB ID 6EFR) with Phaser in Phenix (*Adams et al., 2010*). The experimental phase information of KI-soaked crystals of iNicSnFR3adt was obtained with MR-SAD using AutoSol in Phenix (*Adams et al., 2010*). Molecular replacements of the remaining structures were carried out with the refined model of iNicSnFR3adt. Iterative refinement and model building cycles for all structures were carried out separately with phenix.refine in Phenix (*Adams et al., 2010*) and Coot (*Emsley et al., 2010*).

## Directed evolution of iDrugSnFR proteins using bacterial-expressed protein assays

Starting with iAChSnFR and intermediate biosensor constructs of that sensor, we constructed and optimized iDrugSnFRs for each drug partner during iterative rounds of SSM as previously described (*Bera et al., 2019*; *Shivange et al., 2019*). We utilized the 22-codon procedure including a mixture of three primers, creating 22 unique codons encoding the 20 canonical amino acids (*Kille et al., 2013*). The 22-codon procedure yields an estimated >96% residue coverage for a collection of 96 randomly chosen clones.

A Tecan Spark M10 96-well fluorescence plate reader (Tecan, Männedorf, Switzerland) was used to measure baseline and drug-induced fluorescence ($F_0$ and $\Delta F$, respectively). Bacterial lysates were tested with excitation at 485 nm and emission at 535 nm. Lysates were also measured against

choline to evaluate potential endogenous intracellular binding. Promising clones were amplified and sequenced. The optimally responding construct in each round of SSM was used as a template for the next round of SSM.

S-slope allows for comparison between iDrugSnFRs with differing $\Delta F_{max}/F_0$ values (**Bera et al., 2019**) at the beginning of the dose–response relation, which is usually the pharmacologically relevant range. With lysates or purified protein, which allow complete dose–response relations, the Hill coefficient is near 1.0. We therefore calculated

$$S_{slope} = \frac{\frac{\Delta F_{max}}{F_0}}{EC_{50}},$$

in units of $\mu M^{-1}$.

## Measurements of purified iDrugSnFRs

Biosensors selected for further study were purified using a $His_6$ sequence using an ÄKTA Start FPLC (GE Healthcare, Chicago, IL) as previously described (**Shivange et al., 2019**). Performance of protein quantification and dose–response relations for drug–sensor partners was also as previously described (**Shivange et al., 2019**). Where appropriate, we corrected for depletion of the ligand by binding with the equation,

$$\frac{\Delta F}{\Delta F_{max}} = \frac{k_D + [S] + [L] - \sqrt{([s]^2 + [L]^2 + K_D^2) - 2[S][L] + 2[S]K_D + 2[L]K_D}}{2[s]},$$

where $K_D$ is the ligand–sensor equilibrium dissociation constant (we assume that $K_D = EC_{50}$), [S] is the iDrugSnFR protein concentration (typically 100 nM), and [L] is the nominal ligand concentration.

## Isothermal titration calorimetry

Experiments were performed on an Affinity ITC (TA instruments, New Castle, DE) at 25°C. The iDrugSnFR protein was buffer-exchanged into 3× PBS, pH 7.0. The nicotinic agonists were dissolved in the same buffer. 800 µM cytisine (Sigma-Aldrich, Munich, Germany) was titrated into 80 µM iCytSnFR, 160 µM 10-fluorocytisine was titrated into 16 µM iCyt_F_SnFR. 470 µM 9-bromo-10-ethylcytisine was titrated into 47 µM iCyt_BrEt_SnFR. 1.5 mM dianicline (Tocris, Bio-Techne, Minneapolis, MN) was titrated into 150 µM iDianiSnFR. Analysis, including correction for changes in enthalpy generated from the dilution of the ligands, was performed using a single-site binding model in the manufacturer's NanoAnalyze software.

## Stopped-flow kinetic analysis

Kinetics were determined by mixing equal volumes of 0.2 µM iDrugSnFR protein (in 3× PBS, pH 7.0) with varying concentrations of cognate ligand in an Applied Photophysics (Surrey, UK) SX20 stopped-flow fluorimeter with 490 nm LED excitation and 510 nm long-pass filter at room temperature (22°C). 'Mixing shots' were repeated five times and averaged (except for 100 s experiments, which were collected only once). Standard deviations are not included on the plots, but are nearly the same size as the data markers. The first 3 ms of data were ignored because of mixing artifacts and account for the dead time of the instrument.

Data were plotted and time courses were fitted, when possible, to a single exponential, and $k_{obs}$ was plotted as a function of [ligand]. The linear portion of that graph was fit, with the slope reporting $k_1$ and the y-intercept reporting $k_{-1}$. When the time course did not fit well to a single rising exponential, it was fitted to the sum of two increasing exponentials, and the first rise ($k_{obs1}$) was treated as above to determine $k_1$ and $k_{-1}$.

## Expression in mammalian cells

We constructed two variants of each iDrugSnFR for expression in mammalian cells. The PM (suffix _PM) and ER (suffix _ER) variants were constructed by circular polymerase extension cloning (**Quan and Tian, 2009**). To create the _PM constructs, we cloned the bacterial constructs into pCMV(MinDis), a variant of pDisplay (Thermo Fisher Scientific, Waltham, MA) lacking the hemagglutinin tag (**Marvin et al., 2013**). To generate the _ER constructs, we replaced the 14 C-terminal amino acids (QVDE-QKLISEEDLN, including the Myc tag) with an ER retention motif, QTAEKDEL (**Shivange et al., 2019**).

We transfected the iDrugSnFR cDNA constructs into HeLa and HEK293T cells. Cell lines were purchased from ATCC (Manassas, VA) and cultured according to ATCC protocols. We purchased new aliquots of the cell lines listed above at 6-month intervals to ensure reproducibility. Mycoplasma contamination was assayed at 6-month intervals and was negative over the course of these experiments. For chemical transfection, we utilized either Lipofectamine 2000 or Lipofectamine 3000 (Thermo Fisher Scientific) following the manufacturer's recommended protocol. Cells were incubated in the transfection medium for 24 hr and imaged 24–48 hr after transfection.

## Millisecond timescale microperfusion

HEK293T cells were imaged using a Nikon (Tokyo, Japan) DIAPHOT 300 with a Zeiss ×63 objective (1.5 NA). Because the ligand concentration after micro-iontophoretic drug application (*Shivange et al., 2019*) is unknown, we applied drugs with a laminar-flow microperfusion (model SS-77B Fast-Step perfusion system; Warner Instruments, Holliston, MA). In an array of three square glass capillaries (600 μ i.d.), the center capillary contained vehicle (Hanks buffered salt solution [HBSS]) plus drug, while the two outer capillaries contained vehicle only. Vehicle also flowed from a separate input connected to the bath perfusion system. Solution exchange, measured by loading the center capillary with dye, had a time constant of 90 ± 20 ms (n = 6 trials).

We used Fiji ImageJ and Origin Pro 2018 (OriginLab, Northampton, MA) to fit the rise and decay of the iCytSnFR_PM drug response to the sum of one or two exponential components. An F-test determined whether two exponential components fit the data significantly better than 1 ($p < 0.05$). Statistical comparisons between groups were carried out using ANOVA.

## AAV production and transduction in primary mouse hippocampal neuronal culture

The adeno-associated virus plasmid vector AAV9-hSyn was described previously (*Challis et al., 2019*). Virus was purified using the AAVpro Purification Kit (TakaraBio USA). Mouse embryo dissection and culture were previously described (*Shivange et al., 2019*). About 4 days after dissection, we transduced the _ER construct at an MOI of 0.5–5 × $10^4$; and separately, the _PM construct was transduced at an MOI of 0.5–1 × $10^5$. Neurons were imaged ~2–3 weeks post-transduction.

## Time-resolved fluorescence measurements in live mammalian cells and primary mouse hippocampal neuronal culture

Time-resolved dose–response imaging was performed on a modified Olympus IX-81 microscope (Olympus Microscopes, Tokyo, Japan), in widefield epifluorescence mode using a ×40 lens. Images were acquired at 2–4 frames/s with a back-illuminated EMCCD camera (iXon DU-897, Andor Technology USA, South Windsor, CT), controlled by Andor IQ3 software. Fluorescence measurements at $\lambda_{ex}$ = 470 nm and the epifluorescence cube were as previously described (*Shivange et al., 2019*; *Srinivasan et al., 2011*).

Solutions were delivered from elevated reservoirs by gravity flow, via solenoid valves (Automate Scientific, Berkeley, CA), then through tubing fed into a manifold, at a rate of 1–2 ml/min. The vehicle was HBSS. Other details have been described (*Shivange et al., 2019*; *Srinivasan et al., 2011*). Data analysis procedures included subtraction of 'blank' (extracellular) areas and corrections for baseline drifts using Origin Pro 2018.

## Spinning disk confocal fluorescence images

HeLa cells and mouse primary hippocampal culture were transfected or transduced as described above. Live-cell images were collected using a Nikon Ti-E spinning disk laser scanning confocal inverted microscope equipped with ×100 objective, 1.49 NA (oil), 120 μm WD. The laser wavelength was 488 nm at 15% power. Dishes were imaged in a custom incubator (Okolab, Ottaviano, Italy) at 37°C and 5% $CO_2$. Initial images were taken in HBSS. To add drug, we doubled the bath volume by adding HBSS containing drug using a hand-held pipette. The final drug concentrations were dianicline, 15 μM; cytisine, 10 μM; 10-fluorocytisine, 10 μM; and 9-bromo-10-ethylcytisine, 7.5 μM.

## LogD calculations

We used Chemicalize (https://chemaxon.com/products/chemicalize). The software uses algorithms to calculate logP and $pK_a$. The software then calculates

$$LogD_{pH7.4} = logP -log[1 + 10^{7.4 - pKa}].$$

## Plasmid availability

We will deposit plasmids with the following cDNAs at Addgene:

iDianiSnFR
iCytSnFR
iCyt_F_SnFR
iCyt_BrEt_SnFR

We will deposit the following plasmids at Addgene:

pCMV(MinDis)-iDianiSnFR_PM
pCMV(MinDis)-iCytSnFR_PM
pCMV(MinDis)-iCyt_F_SnFR_PM
pCMV(MinDis)-iCyt_BrEt_SnFR_PM
pCMV(MinDis)-iDianiSnFR_ER
pCMV(MinDis)-iCytSnFR_ER
pCMV(MinDis)-iCyt_F_SnFR_ER
pCMV(MinDis)-iCyt_BrEt_SnFR_ER
pAAV9-hSyn-iDianiSnFR_PM
pAAV9-hSyn-iCytSnFR_PM
pAAV9-hSyn-iCyt_F_SnFR_PM
pAAV9-hSyn iCyt_BrEt_SnFR_PM
pAAV9-hSyn-iDianiSnFR_ER
pAAV9-hSyn-iCytSnFR_ER
pAAV9-hSyn-iCyt_F_SnFR_ER
pAAV9-hSyn iCyt_BrEt_SnFR_ER

## Acknowledgements

We thank Stefan Petrovic for his stewardship of the isothermal titration calorimeter in the Caltech Center for Molecular Medicine, Jens Kaiser for help with structural studies at the Caltech Molecular Observatory, the Gradinaru lab and Caltech CLOVER Center for help with viral vectors, and Andres Collazo and Giada Spigolon at the Caltech Biological Imaging Facility. We thank Zoe Beatty, Kallol Bera, Eve Fine, Shan Huang, Elaine Lin, Stephen Mayo, Lin Tian, and Elizabeth Unger for advice and guidance. We thank Achieve Life Sciences for a gift of cytisine. California Tobacco-Related Disease Research Program (TRDRP) (27 FT-0022), ALN. California Tobacco-Related Disease Research Program (TRDRP) (27IP-0057), HAL. California Tobacco-Related Disease Research Program (TRDRP) (T29IR0455), DAD. NIH (GM-123582, DA043829), HAL. NIH (DA049140, GM7616), AKM. Howard Hughes Medical Institute (LLL, JSM, DCR). UK Engineering and Physical Sciences Research Council (no. EP/N024117/1), TG. Leiden University International Studies Fund (LISF L18020-1-45), LL.

## Additional information

### Funding

| Funder | Grant reference number | Author |
| --- | --- | --- |
| Tobacco-Related Disease Research Program | Postdoctoral Training Fellowship (27FT-0022) | Aaron L Nichols |
| Tobacco-Related Disease Research Program | High Impact Pilot Award (27IP-0057) | Henry A Lester |

| Funder | Grant reference number | Author |
|---|---|---|
| Tobacco-Related Disease Research Program | High Impact Research Project Award (T29IR0455) | Dennis A Dougherty |
| National Institute of General Medical Sciences | Research Project (GM-123582 | Henry A Lester |
| National Institute on Drug Abuse | R21,DA043829 | Henry A Lester |
| Howard Hughes Medical Institute | | Jonathan S Marvin Douglas C Rees Loren L Looger |
| Engineering and Physical Sciences Research Council | No. EP/N024117/1 | Timothy Gallagher |
| Leiden University International Studies Fund | (LISF L18020-1-45) | Laura Luebbert |
| National Institute on Drug Abuse | Exploratory/ Developmental Grants (R21) (DA049140) | Anand K Muthusamy |
| National Institute of General Medical Sciences | Predoctoral Training in Biology and Chemistry (T32) (GM7616) | Anand K Muthusamy |

The funders had no role in study design, data collection and interpretation, or the decision to submit the work for publication.

## Author contributions

Aaron L Nichols, Conceptualization, Data curation, Formal analysis, Funding acquisition, Investigation, Methodology, Supervision, Validation, Visualization, Writing – original draft, Writing – review and editing; Zack Blumenfeld, Data curation, Formal analysis, Investigation, Methodology, Validation, Visualization, Writing – review and editing; Chengcheng Fan, Conceptualization, Data curation, Formal analysis, Investigation, Methodology, Validation, Visualization, Writing – original draft, Writing – review and editing; Laura Luebbert, Bruce N Cohen, Data curation, Formal analysis, Investigation, Methodology, Writing – review and editing; Annet EM Blom, Philip M Borden, Conceptualization, Investigation; Jonathan S Marvin, Data curation, Formal analysis, Investigation, Methodology, Validation, Writing – original draft, Writing – review and editing; Charlene H Kim, Data curation, Investigation, Methodology, Resources, Writing – review and editing; Anand K Muthusamy, Conceptualization, Data curation, Formal analysis, Investigation, Methodology, Resources, Validation, Visualization; Amol V Shivange, Data curation, Validation; Hailey J Knox, Conceptualization, Methodology, Resources, Writing – review and editing; Hugo Rego Campello, Data curation, Resources, Validation; Jonathan H Wang, Investigation, Methodology, Resources; Dennis A Dougherty, Conceptualization, Data curation, Funding acquisition, Project administration, Resources, Supervision; Loren L Looger, Conceptualization, Funding acquisition, Project administration, Resources, Supervision, Writing – review and editing; Timothy Gallagher, Conceptualization, Funding acquisition, Project administration, Supervision, Writing – review and editing; Douglas C Rees, Data curation, Funding acquisition, Project administration, Supervision, Validation, Writing – review and editing; Henry A Lester, Conceptualization, Data curation, Formal analysis, Funding acquisition, Methodology, Project administration, Resources, Software, Supervision, Visualization, Writing – review and editing

## Author ORCIDs

Aaron L Nichols http://orcid.org/0000-0001-9341-0049
Zack Blumenfeld http://orcid.org/0000-0002-4627-5582
Chengcheng Fan http://orcid.org/0000-0003-4213-5758
Laura Luebbert http://orcid.org/0000-0003-1379-2927
Amol V Shivange http://orcid.org/0000-0002-4169-2969
Douglas C Rees http://orcid.org/0000-0003-4073-1185
Henry A Lester http://orcid.org/0000-0002-5470-5255

## Ethics

Generation of primary mouse hippocampal culture was performed in strict accordance with the recommendations in the Guide for the Care and Use of Laboratory Animals of the National Institutes of Health. All of the animals were handled according to the approved institutional animal care and use committee (IACUC) protocols (IA19-1386) of California Institute of Technology. Dissections were performed after euthanasia of the pregnant mouse and every effort was made to minimize suffering.

## Decision letter and Author response

Decision letter https://doi.org/10.7554/eLife.74648.sa1
Author response https://doi.org/10.7554/eLife.74648.sa2

# Additional files

## Supplementary files

- Supplementary file 1. Data collection and refinement statistics of iNicSnFR 3a.
- Supplementary file 2. Stopped-flow model determined rate constants.
- Transparent reporting form

## Data availability

The plasmids and associated database entries are available from Addgene (as named in our manuscript) with genetic maps. The Protein Data Bank has published the crystallographics and structural data (accession codes 7S7T, 7S7U, 7S7V). Supplementary File 1 gives relevant details.

The following datasets were generated:

| Author(s) | Year | Dataset title | Dataset URL | Database and Identifier |
| --- | --- | --- | --- | --- |
| Fan C, Shivange AV, Looger LL, Lester HA, Rees DC | 2021 | iNicSnFR3a Nicotine Sensor comprising Periplasmic Binding sequence plus Fluorescent Sequence with varenicline bound | https://www.rcsb.org/structure/7S7T | RCSB Protein Data Bank, 7S7T |
| Fan C, Shivange AV, Looger LL, Lester HA, Rees DC | 2021 | Crystal structure of iNicSnFR3a Fluorescent Nicotine Sensor with nicotine bound | https://www.rcsb.org/structure/7S7U | RCSB Protein Data Bank, 7S7U |
| Fan C, Shivange AV, Looger LL, Lester HA, Rees DC | 2021 | Crystal structure of iNicSnFR3a Fluorescent Nicotine Sensor | https://www.rcsb.org/structure/7S7V | RCSB Protein Data Bank, 7S7V |

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

## Appendix 1

### iDrugSnFR structural transition

Because all our iDrugSnFRs derive from the *Thermoanaerobacter* sp. X513 choline/betaine binding protein, we asked whether the design process affected conformational changes between the nonfluorescent and fluorescent states. Principal component analysis showed that component 1 accounted for 75% of the total conformational change, which was associated with a hinge-like motion. Component 2 accounted for 17% of the total conformational change, which was associated with the overall conformational difference between the PBPs of iDrugSnFRs and the other published betaine and choline binding proteins. The liganded and apo biosensor structures partitioned into two groups (*Figure 1*). Component 1 accounted for the Venus-flytrap conformational change and was represented best by the structural alignment of varenicline bound to iNicSnFR3adt and the apo iNicSnFR3adt (*Figure 1—figure supplement 1*). As in other betaine or choline binding protein structures in the Protein Data Bank, the liganded conformations differed little from each other, and the conformational change between liganded and apo states related to the Venus-flytrap mechanism. Component 2 accounted for a small fraction of the conformational differences between the native structures and optimized biosensor. This small difference was also observed when the choline-bound iAChSnFRdtbp and *Bacillus subtilis* structures were aligned (RMSD of 1 Å) (*Fan, 2020*).

## Appendix 2

### Further details of kinetic experiments: 'Methods and results'

Millisecond timescale microperfusion with iCytSnFR_PM

#### Cell culture

The iCytSnFR_PM sensor was expressed in HEK293T cells because the high level of protein expression provided by this cell line improved the fluorescent signal/noise ratio. The cells were plated at a density of $1.25 \times 10^5$ cells per dish on gelatin-coated glass coverslips attached to the bottom of a 35 mm culture dish and incubated at 37°C. 24 hr after plating, the cells were transfected with iCytSnFR_PM DNA (200 ng per dish) using Lipofectamine 3000 (Thermo Fisher Scientific). The transfection medium was removed after a 24 hr incubation and replaced with fresh growth medium. We incubated the transfected cells for another 24–72 hr to obtain sufficient iCytSnFR_PM membrane protein density for fluorescent imaging.

#### Fluorescent microscope

A blue LED (488 nm wavelength) provided epifluorescent illumination. The bandwidth and peak of the excitation and emission spectra were 460–500 nm and 480 nm (excitation), and 515–550 nm and 535 nm (emission), respectively (Cat. #41001 FITC filter cube). Immediately prior to imaging, we removed the culture dishes from the incubator and rinsed them three times with HBSS warmed to 37°C. After rinsing, we pried the coverslip off the bottom of the dish with a single-edged razor blade and resealed it to the bottom of a plastic perfusion chamber (model RC-25 diamond shape, Warner Instruments) with silicone grease. At this point, the chamber and coverslip were mounted on a motorized microscope stage and perfused continuously with HBSS at 18°C from a gravity-fed, bath perfusion system to keep the cells oxygenated. The fluid level in the chamber was regulated by suction from a syringe needle.

#### Drug microperfusion

The major components of the microperfusion system were (1) a parallel array of three, square glass capillaries (600 µm i.d.) that delivered the ligand to the cells, (2) a stepping motor that translated the capillary array horizontally above the coverslip surface, and (3) a controller that determined the motion and step size of the motor. Both the array and stepping motor were mounted on a motorized micromanipulator (model MP-285, Sutter Instrument Co., Novato, CA) that allowed us to position it in the chamber. Fluid flowed continuously from all three capillaries into the chamber during the experiment, as well as vehicle from a separate input connected to the bath perfusion system. The center capillary in the array contained vehicle plus drug, while the two outer capillaries contained vehicle only. Outflow from the end capillaries kept ligand flow from the center capillary confined to a laminar stream. We applied drugs to the cells in 5–30 s step-like pulses. The array was positioned so that outflow from an end capillary bathed the cells in the field of view with vehicle initially. To apply the ligand to the cells, we stepped the entire array horizontally by 700 µm so that vehicle plus drug from the center capillary bathed the cells. Ligand application was terminated by returning the array to its initial position.

#### Data acquisition

A computer equipped with the Clampex v.9 software and a Digidata 1200 series interface (Molecular Devices, San Jose, CA) controlled both the timing of ligand application and initiated image acquisition. A camera (model ORCA-3G, Hamamatsu, Hamamatsu City, Japan) attached to a microscope port recorded the images. A second computer equipped with the HCImage V3.0 software and a digital camera interface (Hamamatsu) controlled the camera operation, set the parameters of image acquisition, and stored the recorded image sequences. We recorded image sequences of 170–200 frames at rates of 1–8.9 frames/s to visualize the time course of the iCytSnFR_PM response. Event markers in the image file marked the timing of ligand application. We started acquiring images 5–10 s before applying the ligand to establish a baseline fluorescence and continued for 5–180 s after applying drug to record the decay of the response.

#### Millisecond timescale microperfusion with iCytSnFR_PM (results)

##### Biphasic decay components of the cytisine response

One-way ANOVA showed a significant difference between the relative amplitudes of the decay components after the 5, 8, 10, and 15 µM cytisine pulses (ANOVA, p<0.01, df = 3,32). Post-hoc

comparisons showed that the relative amplitude of the slow component $A_s/(A_f + A_s)$ (where $A_s$ is the amplitude of the slow component and $A_f$ is the amplitude of the fast) of the 5 µM cytisine decay (24% ± 7%, n = 6 areas [13 cells]) was significantly (Tukey tests, p<0.05) less than the mean $A_s/(A_f + A_s)$ of the three other concentrations (8, 10, 15 µM) (66% ± 1%, n = 30 areas [92 cells]). The $A_s/(A_f + A_s)$s after pulses of 8, 10, and 15 µM cytisine did not differ significantly (p>0.05).

## Growth phase of the cytisine response

We visualized the time course of the growth phase of the iCytSnFR_PM response to 1–15 µM cytisine using the Fast-Step microperfusion system (*Appendix 2—figure 1*). The growth phases of responses to 30 s applications of 1–4 µM cytisine were biphasic and fitted best by the sum of two terms describing an exponential rise to a maximum (*Appendix 2—figure 1A*). The range of the faster rate constants ($kf_{on}$s) for 1–4 µM cytisine was 0.41–1.22 s$^{-1}$ (n = 40 areas [138 cells]), and the range for the slower rate constants was 0.01–0.1 s$^{-1}$ (n = 37 areas [128 cells]). The faster component dominated the rising phase of the 1–4 µM cytisine response. The relative amplitude of the faster component $A_f/(A_f + A_s)$ was 77% ± 1%, n = 37 areas (128 cells). It was not affected significantly by cytisine concentration in the 1–4 µM range (ANOVA, df = 3 model, 33 error, p=0.08). The responses to 5–10 s applications of 5–15 µM cytisine reached steady state within a 5 s application. They were fitted adequately by a single negative exponential rise to a maximum (*Appendix 2—figure 1B*). The single rate constant for the rising phase in individual areas ranged from 0.86 to 2.65 s$^{-1}$ (n = 37 areas [133 cells]). We pooled the fast rate constants for the 1–4 µM cytisine responses with the single rate constants for the 5–15 µM responses to obtain an overall [cytisine]-$kf_{on}$ relation for the 1–15 µM cytisine concentration range (*Appendix 2—figure 2A*). Linear least-squares regression showed that the [cytisine]-$kf_{on}$ relation over this concentration range was approximately linear (r = 0.98, n = 8 concentrations, p<0.05) with a slope and intercept of 1.5 ± 0.1 10$^5$ (Ms)$^{-1}$ and 0.43 ± 0.05 s$^{-1}$ (± SE), respectively.

Nevertheless, the [cytisine]-$kf_{on}$ data from the microperfusion experiments deviated from linearity at cytisine concentrations ≥ 8 µM, suggesting that [cytisine]-$kf_{on}$ relation for the microperfusion data was more hyperbolic than linear. Using the slower decay rate constant for cytisine (0.146 ± 0.006 s$^{-1}$) as the value for the $kf_{on}$ at 0 µM cytisine, a hyperbolic relation fitted the cytisine-$kf_{on}$ relation for 0–10 µM cytisine significantly better than a straight line (*Appendix 2—figure 2B*, F-test, p<0.05).

In contrast to the faster rate constant of the rising phase $kf_{on}$, the slower rate constant $ks_{on}$ decreased significantly as the cytisine concentration increased from 1 to 4 µM (r = 0.78, n = 4 concentrations, p<0.05, *Appendix 2—figure 2D*). The slope and intercept of a straight line fit to the [cytisine]-$ks_{on}$ data using least-squares regression were –1.5 ± 0.06 × 10$^4$ M$^{-1}$ s$^{-1}$ and 0.09 ± 0.2 s$^{-1}$ (± SE, n = 4 concentrations), respectively.

## Comments on a kinetic model

Our subsecond data with cytisine at iCytSnFR are the most complete, comprising both stopped-flow and HEK293T cell microperfusion. We therefore discuss the cytisine-iCytSnFR kinetics.

Recent literature on kinetics of PBPs emphasizes a modified induced-fit model with the additional possibility that the apo PBP can also undergo spontaneous activation (termed the 'closed state' in the SBP literature; *de Boer et al., 2019*; *Gouridis et al., 2015*). Such a scheme, shown in *Appendix 2—figure 3*, resembles the three-state model we and colleagues developed to account for iSeroSnFR (*Unger et al., 2020*).

We simulated the scheme using the MATLAB Simbiology toolbox. The following rate constants account for the stopped-flow and millisecond perfusion data within a factor of 3: $k_{bind2}$, 0.5 × 10$^6$ / M$^{-1}$ s$^{-1}$; $k_{unbind2}$, 0.4/ s; $k_{iso(+)}$, 0.001/ s; $k_{iso(-)}$, 0.01/ s; the apo fluorescent state has 0.1 times the brightness of the bound fluorescent state.

The three-state model predicts the experimental observation (*Appendix 2—figure 2D*) that the rate constant of the slower component of the kinetics decreases as the ligand concentration increases. For the cytisine-iCytSnFR case reported in this paper, we conclude that the apo, fluorescent state is less bright than the bound state (shown by the different colors of the cpGFP moiety).

Interestingly, the three-state model fitted the kinetic data best if we assumed that there is a population of higher-sensitivity iDrugSnFRs in HEK293T cells, with an $EC_{50}$ at least 10 times less than we observed with the stopped-flow and HeLa cell data. A more complex, 'square' four-state model, comprising both ligand binding and protein conformational changes, has been applied to equilibrium measurements of cGaMP (*Barnett et al., 2017*). Full kinetic predictions of the four-state model are available (*Lancet and Pecht, 1976*).

The fragmentary kinetic data for acetylcholine at iCytSnFR_PM suggest equilibrium and rate constants in the same broad range as for cytisine. However, the kinetic data for varenicline suggest that ligand unbinding dominates the decay phase, with a rate constant <0.01 s⁻¹. As a caution, recent data show that mechanisms at PBPs (part of the larger class of substrate binding proteins [SBPs]) can change fundamentally with even a single mutation (*Nguyen et al., 2018*). We therefore wish to avoid generalizing past the single iDrugSnFR we consider here.

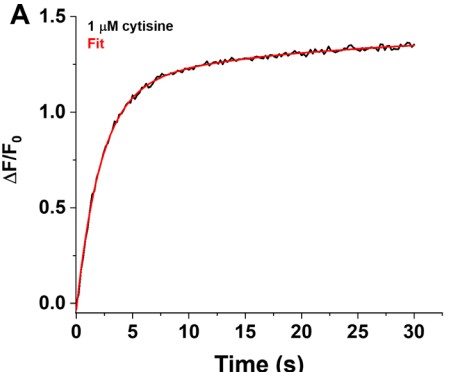 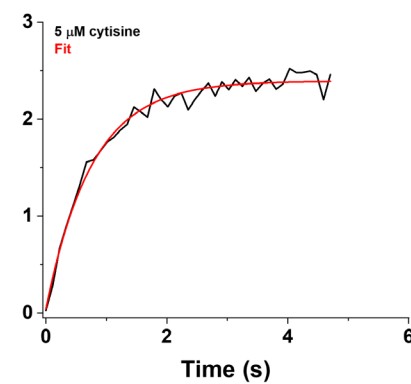

**Appendix 2—figure 1.** Rising phase of the iCytSnFR_PM response to cytisine in HEK293T cells. (**A**) Example of the biphasic rising phase of a 1 µM cytisine response in an individual area (black trace, mean of four cells). Cytisine was applied for 30 s. The fast and slow time constants of the rising phase ($\tau f_{0n}$, $\tau s_{0n}$) were 2.15 ± 0.05 s and 17 ± 3 s (n = 151 frames, sampling rate of 5 frames/s), respectively. The red line is a fit to the sum of two declining exponentials ($R^2$ of 0.998). It was significantly better than that to a single negative exponential rise to maximum component and a constant term (F-test, p<0.05). The $A_s/(A_s + A_f)$ was 20%. (**B**) Example of the rising phase of a response to a 5 s application of 5 µM cytisine in an individual area (mean of 10 cells, two replicates). The response appeared to be monophasic with a single time constant ($\tau_{0n}$) of 0.76 ± 0.04 s (n = 43 frames, sampling rate of 9.8 frames/s). The red line is a fit to the sum of a negative exponential rise to a maximum, and constant, term ($R^2$ of 0.98).

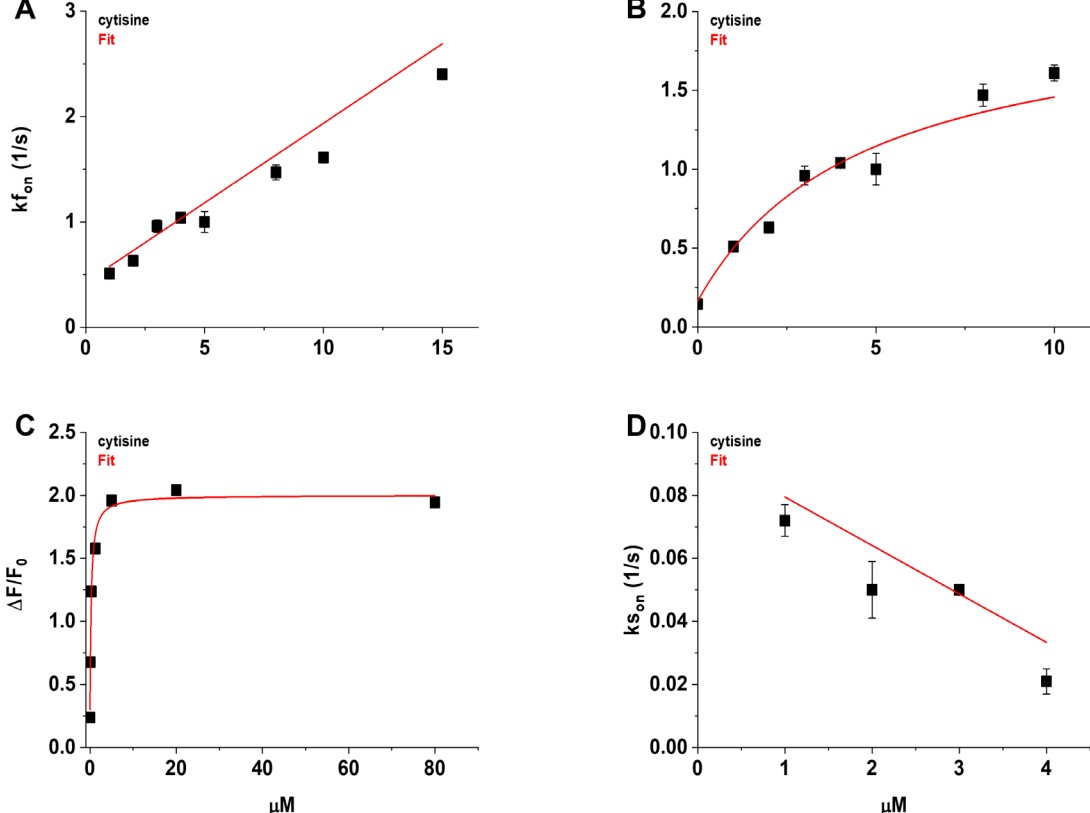

**Appendix 2—figure 2.** Concentration dependence of the fast ($kf_{on}$) and slow rising rate constants ($ks_{on}$) of the cytisine response. (**A**) The [Cytisine]-$kf_{on}$ relation was approximately linear between 1 and 15 µM cytisine. Symbols (filled squares) are the mean $kf_{on}$ for the individual cytisine concentrations tested (n = 7–10 areas per concentration, 29–40 cells). The red line is a regression line fit to the data using linear least-squares regression. See text for the values of the slope, intercept, and correlation coefficient. (**B**) The [Cytisine]-$kf_{on}$ relation for the $kf_{on}$ was more hyperbolic than linear between 0 and 10 µM cytisine. Red line is a fit to the sum of a hyperbolic, and constant, term using nonlinear least-squares regression. See text for fitted parameters. We used the mean slow decay rate constant ($ks_{off}$) of the cytisine response for the $kf_{on}$ at 0 µM cytisine (0.146 ± 0.006 s$^{-1}$). (**C**) Concentration–response (CR) relation for the mean steady-state response to cytisine of iCytSnFR_PM sensors expressed in HeLa cells (n = 11 cells, see *Figure 7B*). Red line is the fit to the sum of a hyperbolic, and constant, terms using nonlinear least-squares regression. See text for fitted parameters. Symbols (filled squares) are the mean values for the final 10 s of the steady-state cytisine response. (**D**) The [Cytisine]-$ks_{on}$ relation for 1–4 µM cytisine. Red line is a regression line fit to the data using linear least-squares regression. See text for the slope, intercept, and correlation coefficient. Symbols (filled squares) are the mean $ks_{on}$ for the individual cytisine concentration tested (n = 8–10 areas per concentration, 31–38 cells). Error bars in panels (**A–D**) are ± SEM. Symbols obscure the bars at some concentrations in panels (**A**), (**B**), and (**D**), and all concentrations in panel (**C**).

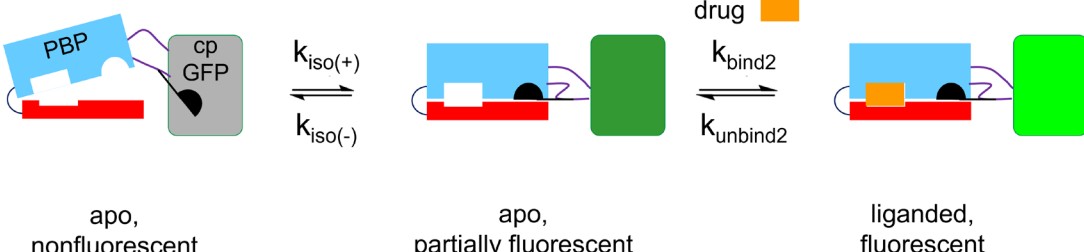

**Appendix 2—figure 3.** A three-state kinetic scheme for iCytSnFR. The diagram contains cartoons of the PBP moiety (blue and red), the linkers (black lines), the Glu78 'candle snuffer' attached to Linker 1 (black), and the cpGFP moiety (gray, dark green, or green). We postulate that the intensity-based drug-sensing fluorescent reporter
*Appendix 2—figure 3 continued on next page*

*Appendix 2—figure 3 continued*

(iDrugSnFR) exists in an apo nonfluorescent state and an apo fluorescent state; these states interconvert with time constants of tens of seconds $k_{(iso(+)}$, $k_{iso(-)})$. Cytisine binds to the apo fluorescent state ($k_{bind2}$), inducing an additional fluorescent state on a briefer time scale. The initial fluorescence increase represents the binding-induced increase, and the slower increase is governed by partial re-equilibration of the two apo states. Upon removal of cytisine after just a few seconds of perfusion (*Figure 6*), the fluorescence decay represents the dissociation of cytisine ($k_{unbind2}$). This scheme resembles the model we and colleagues developed to account for iSeroSnFR (*Unger et al., 2020*). For the iDrugSnFRs reported in this paper, we conclude that the apo, fluorescent state is less bright than the bound state (shown by the different colors of the cpGFP moiety).

