## [Editor Report]

Nichols et al. developed and characterized the first fluorescent sensors for several nicotinic receptor partial agonists relevant to smoking cessation. It is potentially a major advance for the field. They leveraged crystallography to understand the mechanism by which the ligands enhance fluorescence, then characterized top sensors for sensitivity, selectivity, and kinetics, and their utility in plasma membrane and ER sensing in neurons and cell lines. The tools developed by this team will enable investigators to track nicotinic receptor partial agonists in different subcellular compartments with relatively fast time resolution.

---

## [Decision Letter]

**Decision letter after peer review:**

Thank you for submitting your article "Fluorescence Activation Mechanism and Imaging of Drug Permeation with New Sensors for Smoking-Cessation Ligands" for consideration by *eLife*. Your article has been reviewed by 3 peer reviewers, and the evaluation has been overseen by a Reviewing Editor and Kenton Swartz as the Senior Editor. The following individuals involved in review of your submission have agreed to reveal their identity: Ryan E Hibbs (Reviewer #1); Thomas E. Hughes (Reviewer #2); Owen McManus (Reviewer #3).

Congratulations on an excellent piece of work!

Essential revisions:

Please make changes to the text for clarity and answer the individual small questions raised in the reviewers' recommendations below. Also, several reviewers agreed that the impact and interest of the paper for the audience could be broadened by adding a little material to the discussion about how analogous tools may be useful beyond the explicit case of nicotine addiction.

*Reviewer #2 (Recommendations for the authors):*

Great read, this is a beautiful piece of work. My only concern is that the paper covers such a broad swath of disciplines that it might be better with additional background? Biosensor people know remarkably little about target engagement issues in drug discover, pharmacologists are only vaguely aware of biosensors, and medicinal chemists are on a different planet. Is there a way to frame the bigger picture beyond nicotine addiction? It's so easy to see how this sort of approach could be used in SAR in different drug discovery campaigns for example.

---

## [Author Response]

Essential revisions:Please make changes to the text for clarity and answer the individual small questions raised in the reviewers' recommendations below. Also, several reviewers agreed that the impact and interest of the paper for the audience could be broadened by adding a little material to the discussion about how analogous tools may be useful beyond the explicit case of nicotine addiction.

The revised Discussion adds the explicit statement:

“The iDrugSnFR paradigm will be useful beyond the explicit case of nicotine addiction, with application to other exogenous neural drugs.”

Reviewer #2 (Recommendations for the authors):Great read, this is a beautiful piece of work. My only concern is that the paper covers such a broad swath of disciplines that it might be better with additional background? Biosensor people know remarkably little about target engagement issues in drug discover, pharmacologists are only vaguely aware of biosensors, and medicinal chemists are on a different planet. Is there a way to frame the bigger picture beyond nicotine addiction? It's so easy to see how this sort of approach could be used in SAR in different drug discovery campaigns for example.

Amusingly, none of the authors have ever taken a formal course in the disciplines noted: biosensors, drug discovery, pharmacology, medicinal chemistry, or (not mentioned) pharmacokinetics. Lightheartedly, we note that none of us are known to suffer from imposter syndrome.

More seriously, we have presented 2021 SfN abstracts on related biosensors for opioids, SSRIs, and rapidly acting antidepressants. When those papers are submitted and published, we hope to verify Reviewer 2’s prediction. Meanwhile, the revised Discussion ends with the Editor’s suggested statement:

“The iDrugSnFR paradigm will be useful beyond the explicit case of nicotine addiction, with application to other exogenous neural drugs.”